# Wnt- and glutamate-receptors orchestrate stem cell dynamics and asymmetric cell division

Sergi Junyent[†], Joshua C Reeves[†], James LA Szczerkowski, Clare L Garcin, Tung-Jui Trieu, Matthew Wilson, Jethro Lundie-Brown, Shukry J Habib*

Centre for Stem Cells and Regenerative Medicine, King's College London, London, United Kingdom

**Abstract** The Wnt-pathway is part of a signalling network that regulates many aspects of cell biology. Recently, we discovered crosstalk between AMPA/Kainate-type ionotropic glutamate receptors (iGluRs) and the Wnt-pathway during the initial Wnt3a-interaction at the cytonemes of mouse embryonic stem cells (ESCs). Here, we demonstrate that this crosstalk persists throughout the Wnt3a-response in ESCs. Both AMPA and Kainate receptors regulate early Wnt3a-recruitment, dynamics on the cell membrane, and orientation of the spindle towards a Wnt3a-source at mitosis. AMPA receptors specifically are required for segregating cell fate components during Wnt3a-mediated asymmetric cell division (ACD). Using Wnt-pathway component knockout lines, we determine that Wnt co-receptor Lrp6 has particular functionality over Lrp5 in cytoneme formation, and in facilitating ACD. Both Lrp5 and 6, alongside pathway effector β-catenin act in concert to mediate the positioning of the dynamic interaction with, and spindle orientation to, a localised Wnt3a-source. Wnt-iGluR crosstalk may prove pervasive throughout embryonic and adult stem cell signalling.

**\*For correspondence:**
Shukry.habib@kcl.ac.uk

[†]These authors contributed equally to this work

**Competing interests:** The authors declare that no competing interests exist.

## Introduction

Cell-to-cell communication is fundamental to multicellular life. Developmental signals critical for tissue patterning and formation are traditionally considered as diffuse signals that pervade the extracellular matrix. However, signals have often been found to be delivered to target cells or received from source cells via specialized signalling filopodia — cytonemes (*Kornberg and Roy, 2014*).

Stem cells, both embryonic and adult, regulate tissue formation. Many stem cells rely on Wnts as signals for self-renewal and differentiation in the balance of cell fate determination during development, tissue homeostasis, and disease (*Garcin and Habib, 2017*). Wnt ligands bind receptor Fzd, alongside co-receptors Lrp5 or Lrp6, ultimately leading to β-catenin-dependent transcription of target genes. In several systems (*Bertrand, 2016*; *Goldstein et al., 2006*; *Huang and Niehrs, 2014*; *Kaur et al., 2020*; *Sugioka et al., 2011*; *Walston et al., 2004*), including mouse embryonic stem cells (ESCs) (*Habib et al., 2013*), localised Wnts induce oriented asymmetric cell division (ACD).

As cultured single cells that retain the capacity to self-renew and differentiate to all adult lineages, ESCs are an ideal experimental system for studying cell fate determination (*Martello and Smith, 2014*). By combining ESCs and trophoblast stem cells (TSCs) to generate 3D structures that resemble the developing embryo, we recently identified ESC-generated cytonemes that mediate the interaction with Wnt-secreting TSCs (*Junyent et al., 2021*; *Junyent et al., 2020*). To overcome the multiplicity of signals and the difficulty in visualising secreted Wnt ligands, we used Wnt3a tethered to microbeads, showing that ESC-cytonemes selectively recruit self-renewal-promoting Wnt3a ligands. Cytonemes interacting with localised Wnt3a signals engage not only Wnt-receptors, but also ionotropic AMPA/Kainate-type glutamate receptors (iGluRs) (*Junyent et al., 2021*;

*Junyent et al., 2020*). Consequently, rapid calcium (Ca$^{2+}$) transients are generated at the cytoneme, leading to Wnt3a reactivity: Wnt3a-bead recruitment to the cell body (soma), activating Wnt/β-catenin signalling and promoting asymmetric cell division (ACD) of markers of cell fate (*Habib et al., 2013*). iGluRs are primarily studied in excitatory glutamatergic synapses between neurons, however they are increasingly being found in other cell types, such as stem cells (*Genever et al., 1999*; *Morhenn et al., 1994*; *Skerry and Genever, 2001*) and functioning in *Drosophila* development (*Huang et al., 2019*). AMPA- and Kainate-type iGluRs are tetrameric glutamate-sensitive (amongst other ligands *Traynelis et al., 2010*) ion channels. AMPA/Kainate receptors require subunits GriA3 and GriA4, or GriK1 and GriK3 to function, respectively (*Cull-Candy et al., 2006*; *Lerma and Marques, 2013*). Activation increases their ion permeability, inducing rapid ion influxes critical to signal transmission.

However, it remained unknown how the interconnected roles of Wnt-pathway and AMPA/Kainate receptor components influence highly dynamic Wnt3a interactions, and the subsequent Wnt3a-mediated ACD of ESCs.

Here, we resolve the dynamics of localised Wnt3a ligand (Wnt3a-bead) recruitment by single ESCs into three stages: initial cytoneme-mediated interactions; Wnt3a-bead interaction with the cell membrane; and the subsequent effects of the Wnt3a-bead recruitment on spindle orientation and cell fate determination. We generated ESCs knocked-out (KO) for the AMPA- and Kainate-type iGluRs by double knock-out (dKO) of critical protein subunits (AMPA: GriA3/4dKO; Kainate: GriK1/3dKO). Multiple clonal cell lines were investigated alongside pharmacological inhibition, both previously- and newly-generated mutants for Wnt co-receptors (Lrp5KO, Lrp6KO and Lrp5/6dKO *Junyent et al., 2020*, and inducible β-catenin KO ESCs *Raggioli et al., 2014*).

We show that while Lrp6, β-catenin and AMPA/Kainate receptors are required for the population-wide cytoneme-mediated uptake of Wnt3a-beads, neither AMPA nor Kainate receptors are important to the formation of cytonemes themselves. Lrp5, conversely, has an inhibitory role in regulating cytoneme generation in single ESCs and Wnt3a-bead accumulation at cell-population level. In WT ESCs, Wnt3a ligands are recruited from initial cytoneme interaction site at the cell periphery, towards the centre of the cell soma, a process which requires Lrp5, Lrp6, β-catenin, AMPA, and Kainate receptors. Upon progression to mitosis, WT ESCs orient the spindle towards Wnt3a-beads but not inert (chemically inactivated) iWnt3a-beads. Cells lacking Lrp5, Lrp6, β-catenin, AMPA, or Kainate receptors have significantly impaired ability to orient the spindle towards the Wnt3a signal source.

WT ESCs divide asymmetrically in response to a Wnt3a-bead, producing a self-renewed proximal ESC cell and a distal cell that is prone to differentiation (*Habib et al., 2013*). Specifically, we demonstrate that Lrp5 is not required for Wnt3a-mediated ACD, while Lrp6 and β-catenin are crucial. Importantly, we distinguish that AMPA receptors, but not Kainate receptors, are critical in the cell fate determination of Wnt3a-mediated ACD in ESCs.

Taken together, we show that AMPA/Kainate-type iGluR activity is incorporated throughout the interaction of ESCs with localised Wnt3a signals. We dissect the dynamic interaction of Wnt3a-ligand with ESC, from initial uptake to cellular division, showing that its efficient regulation requires Wnt co-receptors and downstream β-catenin in concert with AMPA/Kainate receptor activity. Our multiple clone approach to component KOs suggests that the downstream cellular responses to localised Wnt3a of spindle orientation and asymmetric cell division are impervious to clonal variation in the initial dynamics of the Wnt3a-interaction. Through this, we specify novel differences between the roles of Lrp5 and 6, and between AMPA- and Kainate-type iGluRs in the response to localised Wnt3a.

## Results

### Wnt co-receptors and β-catenin regulate Wnt3a-bead accumulation by modifying the cytonemes of ESCs

We aimed to investigate how components of the Wnt/β-catenin pathway (*Figure 1A*) affect the cellular dynamics of ESCs and their capacity to respond to localised Wnt3a ligands. We used our previously generated ESC lines knocked-out (KO) for the co-receptors of the Wnt/β-catenin pathway: Lrp5 (Lrp5KO), Lrp6 (Lrp6KO) or both Lrp5 and 6 double KO (Lrp5/6dKO) (described in *Junyent et al., 2020*, here named KO #1). An additional clone of each line was newly generated to

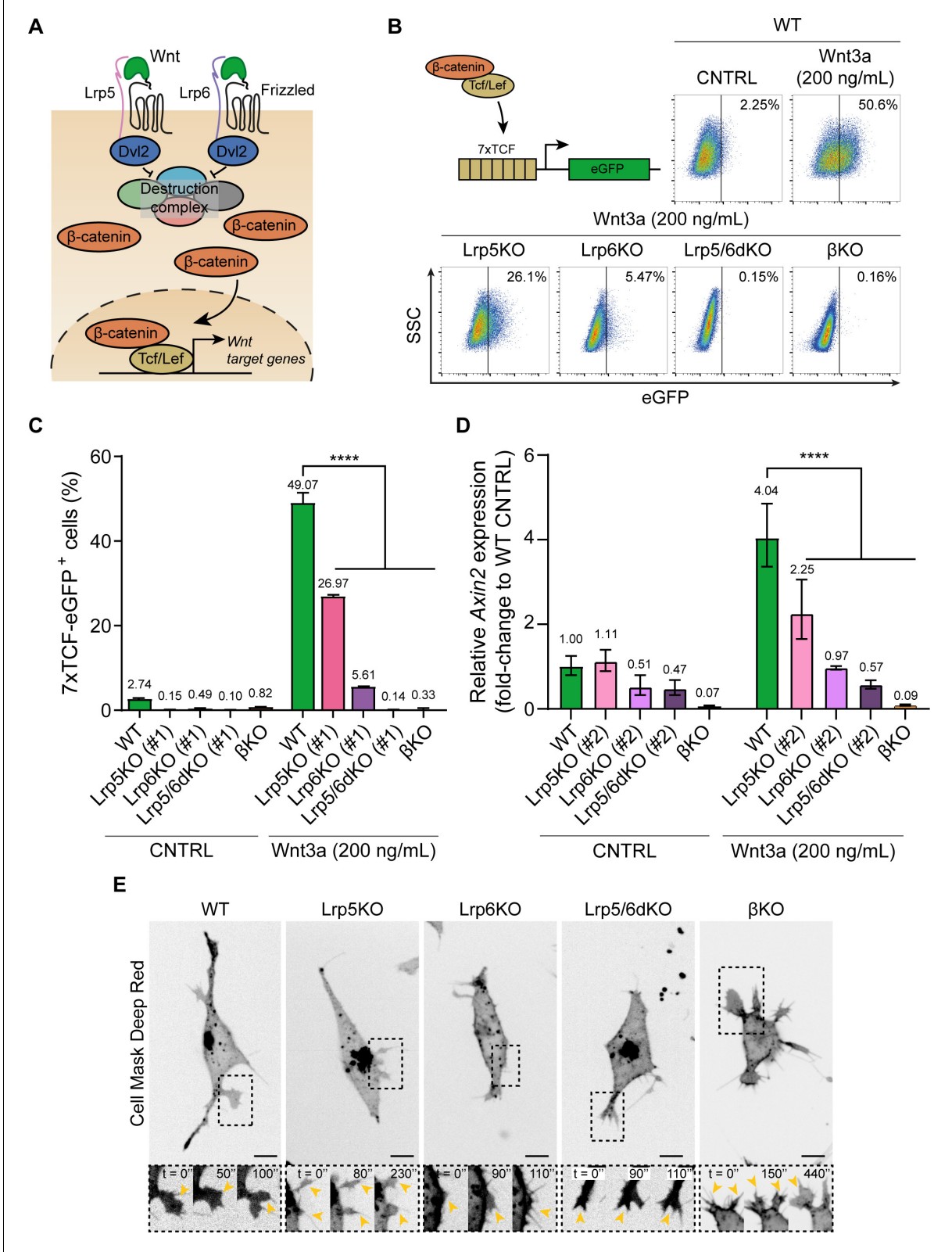

**Figure 1.** Wnt3a responsiveness and formation of dynamic cytonemes in Wnt/β-catenin pathway KO ESCs. (**A**) Simplified schematic representation of the Wnt/β-catenin pathway. Wnt3a binding to the receptor Frizzled and the co-receptors Lrp5 and Lrp6 induces the inhibition of the β-catenin destruction complex (Axin2, Gsk3, Ck1, APC) through Dvl. β-catenin is stabilised and translocated to the nucleus, where it binds Tcf/Lef to initiate the transcription of Wnt3a target genes. (**B**) (*Top-left*) Schematic representation of the 7xTCF-eGFP Wnt/β-catenin reporter. (*Top-right* and *Bottom*)

*Figure 1 continued on next page*

**eLife** Research article

Cell Biology | Stem Cells and Regenerative Medicine

*Figure 1 continued*

Representative dot-plots depicting the levels of eGFP expression in WT or Wnt pathway KO ESCs carrying the 7xTCF-eGFP reporter and treated with CNTRL solution or 200 ng/mL soluble Wnt3a for 24 hr. Y axis is SSC. Values in plots are percentage of GFP+ cells. (C) Percentage of 7xTCF-eGFP+ cells measured by FACS, as described in (B). n = 6 for WT, n = 3 for KO ESCs. Bars and numbers are mean, error bars indicate SEM. Stars indicate statistical significance calculated by two-way ANOVA with Dunnett's multiple comparison test: ****$p<0.0001$. (D) Relative *Axin2* expression in WT or Wnt pathway KO ESCs treated with CNTRL media or 200 ng/mL soluble Wnt3a for 24 hr, measured by qPCR. n = 3. Bars and numbers are geometric mean, error bars indicate geometric SD. Stars indicate statistical significance calculated by two-way ANOVA with Dunnett's multiple comparison test: ****$p<0.0001$. (E) Representative images of the cytonemes in WT, Lrp5KO (#1), Lrp6KO (#1), Lrp5/6dKO (#1) and βKO ESCs. Cells were stained with CellMaskDeep Red and imaged for 10 min every 10 s on a spinning disk confocal, with a 40x/0.95 air objective (incubation at 37°C, 5% $CO_2$). Insets (*below*) are magnifications of black dashed boxes, contrast-enhanced for clarity. Yellow arrowheads indicate thin protrusions. Time is in s. Scale bars are 10 μm. Numeric data used in the figure can be found in the Source Data file.

The online version of this article includes the following source data and figure supplement(s) for figure 1:

**Source data 1.** Numeric data used in *Figure 1*.
**Figure supplement 1.** Genotype of Wnt co-receptor KO ESCs.
**Figure supplement 2.** Representative images of cytonemes in Wnt co-receptor KO ESCs.

investigate the influence of clonal variation during the interaction with Wnt3a signals (named KO #2, *Figure 1—figure supplement 1*). We also used an ESC line with a conditional KO of the pathway effector, β-catenin (βKO) (generated as described in *Junyent et al., 2020*; *Raggioli et al., 2014*).

To validate the response of the clones of the KO cell lines to exogenous Wnt3a ligands, we used cells harbouring the Wnt/β-catenin pathway reporter, 7xTCF-eGFP, in which eGFP is expressed upon pathway activation (*Fuerer and Nusse, 2010*; *Figure 1B and C*). Similar to our previous findings (*Junyent et al., 2020*), in the presence of solubilised Wnt3a ligands 49.07% of wild-type (WT) cells express eGFP, while only a basal level of pathway activation is observed in these cells in the absence of Wnt3a (*Figure 1C*). In comparison, Lrp5KO ESCs have a significant reduction in pathway activation in response to Wnt3a (26.97% eGFP+ cells). In Lrp6KO ESCs, pathway activation is further compromised (5.61% eGFP+ cells). Lrp5/6dKO and βKO ESCs cannot activate the Wnt/β-catenin pathway in response to solubilised Wnt3a addition (*Figure 1B and C*). We also measured the expression of *Axin2* (a universal Wnt/β-catenin target gene) in WT or KO ESCs stimulated with soluble Wnt3a. Again, this demonstrated a reduction in Wnt/β-catenin pathway activation in Lrp5KO and Lrp6KO ESCs, and the absence of responsiveness to Wnt3a in Lrp5/6dKO or βKO cells (*Figure 1D*).

At the single-cell level, ESCs use dynamic cytonemes to contact and recruit Wnt3a ligands immobilised to a microbead (Wnt3a-beads) (*Junyent et al., 2021*; *Junyent et al., 2020*). These cytonemes are shorter in βKO and Lrp6-mutant ESCs (Lrp6KO, Lrp5/6dKO) than in WT ESCs, and are presented in greater numbers in βKO cells (*Junyent et al., 2020*) (exemplified in *Figure 1E*, *Figure 1—figure supplement 2*, *Videos 1–5*).

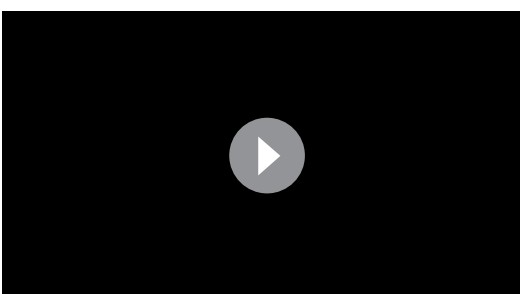

**Video 1.** WT ESC cytonemes. High-quality images of the dynamic cytonemes of WT ESCs. Cells were stained with CellMask Deep Red and imaged in a spinning disk confocal microscope every 10 s, for 10 min. Images are presented as inverted greyscale. Yellow boxes are magnifications, contrast-enhanced for clarity. Time is min and s. Scale bar, 10 μm.

https://elifesciences.org/articles/59791#video1

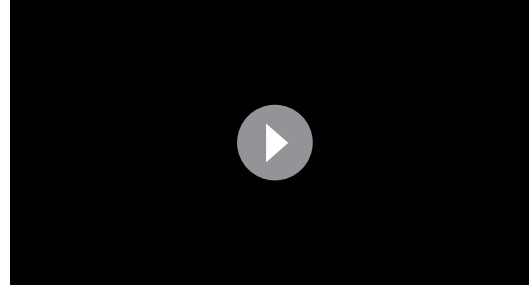

**Video 2.** Lrp5KO ESC cytonemes. High-quality images of the dynamic cytonemes of Lrp5KO ESCs (clone #1 and #2). Cells were stained and imaged as described for *Video 1*. Images are presented as inverted greyscale. Yellow boxes are magnifications, contrast-enhanced for clarity. Time is min and s. Scale bar, 10 μm.

https://elifesciences.org/articles/59791#video2

To investigate how differences in Wnt3a-response and single-cell behaviours affect the accumulation of Wnt3a-beads at the population level over a prolonged period, we cultured ESCs at low density and presented them with dispersed Wnt3a-beads. In some conditions, we presented the cells with inactivated Wnt3a-beads (iWnt3a-beads), dithiothreitol (DTT)-treated to denature the Wnt3a protein while covalently bound to the bead. Cells were then live-imaged each minute for a total of 12 hr and the percentage of cells with cytonemes as well as the proportion of cells contacting Wnt3a-beads were analysed.

WT ESCs begin forming cytonemes upon substrate attachment: within 4 hr, over 40% of WT ESCs had cytonemes (*Figure 2A*). This was coupled with a linear increase in the proportion of WT ESCs with Wnt3a-beads in the first 4 hr of imaging, rising to 54.1% by 4 hr, and 63.5% by 11 hr (*Figure 3A*). ESC cytonemes are ligand-selective and can efficiently recruit localised Wnt3a signals but not iWnt3a-beads (*Junyent et al., 2020*). Indeed, the percentage of WT ESCs with iWnt3a-beads did not rise and remained less than 20% for most of the time period (*Figure 3A*).

Interestingly, >56% of Lrp5KO ESCs generate cytonemes in 3 hr, more than WT ESCs (*Figure 2A*). These cells retained the ability to quickly accumulate Wnt3a-beads, reaching ~50% cells with beads after 3 hr and an overall mean of 61.5% after 6 hr (*Figure 3B*). The rate of increase in the percentage of Lrp6KO (*Figure 2B*), Lrp5/6dKO (*Figure 2C*) and βKO ESCs (*Figure 2D*) with cytonemes is largely similar to WT, with some clonal variability in the Lrp6KO cells (*Figure 2B*). Cytonemes of single ESCs lacking Lrp6 (Lrp6KO, Lrp5/6dKO) exhibit reduced reactivity (decreased Wnt3a-bead uptake following initial contact, *Junyent et al., 2020*). Indeed, Lrp6KO and Lrp5/6dKO showed long-term impaired Wnt3a-bead accumulation at the population level, remaining at ~33% cells with beads over the course of 12 hr. This resembled the interaction of WT cells with iWnt3a-beads (*Figure 3C, D and F*). While cells lacking β-catenin (βKO) retain the capacity to efficiently recruit Wnt3a signals in single-cell–single-bead interactions (*Junyent et al., 2020*), they had significantly impaired long-term Wnt3a-bead accumulation at the population level, reaching a maximum mean of 32.7% cells with beads after 12 hr (*Figure 3E and F*).

Together with cytoneme formation, cell motility provides a mechanism to encounter and interact with the local environment. To understand whether cytoneme formation or Wnt3a-bead accumulation is coupled to cellular motility, we calculated the Mean Squared Displacement per minute (MSD), the average movement per minute, of single cells. Cell motility in WT ESCs is minimal, with a median MSD of 1.08 µm²/min (*Figure 2E*). No significant differences to WT were observed between receptor or β-catenin KO ESCs, with the median MSD for each cell type falling between 0.74 and 1.36 µm²/min (*Figure 2E*).

Altogether, we observed that intrinsic Wnt/β-catenin components control the generation and function of cytonemes without affecting overall cellular motility. Single-cell effects, such as shortening of the cytonemes (in βKO ESCs) or changes in the functionality of the cytonemes (Lrp6KO, Lrp5/6dKO ESCs, or WT ESCs contacting iWnt3a-beads) are reflected in population-level changes in the rate of Wnt3a signal accumulation over a prolonged period.

Despite these differences, it is important to note that, in all conditions, some cells do take up and retain Wnt3a-beads, allowing further analysis of the interaction with localised Wnt3a.

## Wnt3a-bead positioning at the membrane is controlled by Wnt co-receptors and β-catenin

Cytoneme-mediated Wnt3a recruitment is the first stage in the process of signal reception, followed by a period of dynamic cell-signal interaction at the membrane of ESCs. Unless interrupted by the loss of contact (Wnt3a-bead dropping) (*Figure 4A*), this interaction often persists until cell division. The positioning of the signal-receptor complexes on the plasma membrane has been shown to affect signalling activity (*Rangamani et al., 2013*; *Schmick and Bastiaens, 2014*). The Wnt3a-bead approach provides a unique opportunity to quantify the dynamics and positioning of localised Wnt3a on the cell membrane, as well as the roles of the Wnt pathway co-receptors and β-catenin in its regulation. Therefore, we next analysed the interaction of Wnt3a-beads and ESCs after initial cytoneme-mediated recruitment.

We followed cells that recruit a Wnt3a-bead with a cytoneme for 3 hr (180 min) after Wnt3a-bead recruitment, a time point sufficient to measure the kinetics of initial cell-signal interaction, and activate the downstream β-catenin related machinery of the pathway (*Hernández et al., 2012*; *Video 6*).

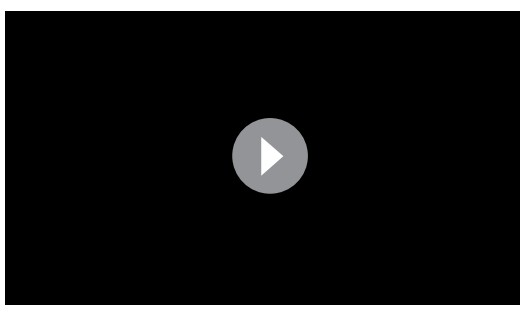

**Video 3.** Lrp6KO ESC cytonemes. High-quality images of the dynamic cytonemes of Lrp6KO ESCs (clone #1 and #2). Cells were stained and imaged as described for *Video 1*. Images are presented as inverted greyscale. Yellow boxes are magnifications, contrast-enhanced for clarity. Time is min and s. Scale bar, 10 µm.

https://elifesciences.org/articles/59791#video3

We measured the distance between the Wnt3a-bead and the cell centre, reporting it as a proportion of the cell radius (*Figure 4B*).

WT ESCs initially recruit Wnt3a-beads with a cytoneme, and rapidly relocate them from the tip to the membrane at the base (between 0.7 and 1.0 of the cell radius), then closer to the cell centre (between 0.3 and 0.7 of the cell radius). Within 1 hr, Wnt3a-beads were, on average, at ~0.5 of the cell radius (*Figure 4B and C*, *Figure 4—figure supplement 1*, *Video 6*). iWnt3a-beads are also initially moved from the WT ESC cytoneme tip to the periphery (0.7–1.0 of the cell radius). However, iWnt3a-beads are not positioned further towards the cell centre, reaching at closest ~0.7 of the cell radius, on average (*Figure 4B and C*).

ESCs lacking Lrp6 (Lrp6KO, Lrp5/6dKO) or β-catenin are likewise, on average, largely unable to position the Wnt3a-bead closer to the cell body (*Figure 4B and E–G*, *Video 6*). In these conditions the bead is stabilised after 90 min in mean positions ranging from 0.62 to 0.70 for Lrp6KO and Lrp5/6dKO, to ~0.70 for βKO ESCs. Lrp5KO ESCs position the Wnt3a-beads at approximately 0.6 after 120 min, further from the centre of the cell than WT ESCs (*Figure 4D*), with clonal differences in the initial positioning.

The dynamics of Wnt3a-cell interaction are also defined by the speed of Wnt3a-bead movement relative to the cell. Thus, we also calculated the MSD of the bead relative to the centre of the cell (*Figure 4H*). iWnt3a-beads held by WT ESCs have a higher rate of movement than Wnt3a-beads, with a median MSD of 4.02 µm²/min versus 3.08 µm²/min for Wnt3a-beads (*Figure 4I*). In mutant ESCs contacting Wnt3a-beads, differences in Wnt3a-bead movement are largely clone-dependent and independent of the genotype of the cell, with the median MSD ranging between 4.25 µm²/min in Lrp5KO (#1) and 2.61 µm²/min in Lrp5/6dKO (#2) (*Figure 4I*).

Finally, we investigated whether positioning of the Wnt3a-bead further from the cell centre led to shorter Wnt3a-cell interactions. We observed that, within the first 3 hr of interaction following cytoneme-recruitment, WT ESCs drop iWnt3a-beads at a significantly higher rate than Wnt3a-beads (44.4% versus 4.08%, *Figure 4J*). However, most mutant ESCs that succeed in recruiting Wnt3a-beads retain them, with rates ranging between

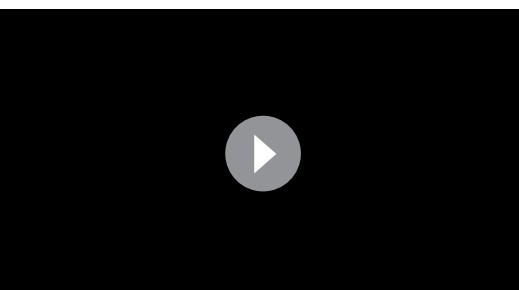

**Video 4.** Lrp5/6dKO ESC cytonemes. High-quality images of the dynamic cytonemes of Lrp5/6dKO ESCs (clone #1 and #2). Cells were stained and imaged as described for *Video 1*. Images are presented as inverted greyscale. Yellow boxes are magnifications, contrast-enhanced for clarity. Time is min and s. Scale bar, 10 µm.

https://elifesciences.org/articles/59791#video4

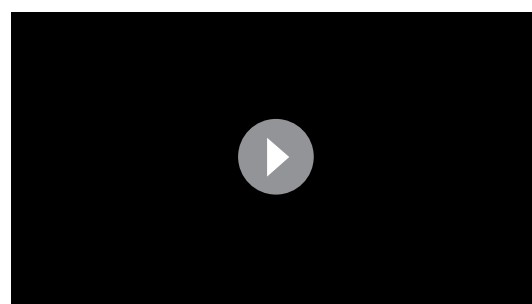

**Video 5.** βKO ESC cytonemes. High-quality images of the dynamic cytonemes of βKO ESCs. Cells were stained and imaged as described for *Video 1*. Images are presented as inverted greyscale. Yellow boxes are magnifications, contrast-enhanced for clarity. Time is min and s. Scale bar, 10 µm.

https://elifesciences.org/articles/59791#video5

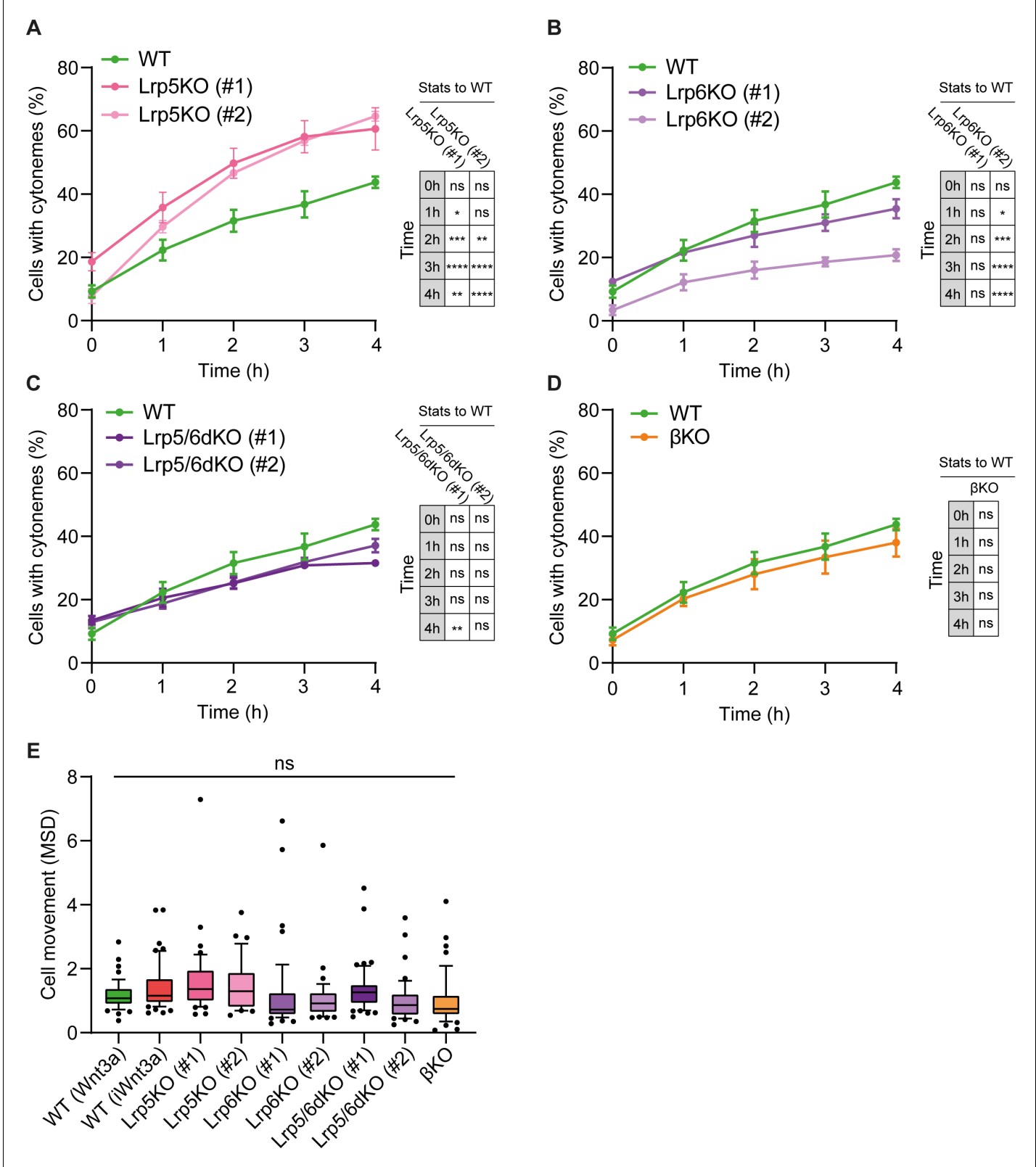

**Figure 2.** Lrp5 controls the generation of cytonemes by ESCs. (A – D) Percentage of WT, Lrp5KO (clone #1 and #2) (A), Lrp6KO (#1 and #2) (B), Lrp5/6dKO (#1 and #2) (C) and βKO ESCs (D) with cytonemes at 0–4 hr after seeding. Line and dots are mean, error bars are SEM. N ≥ 33 cells, n ≥ 3 experiments. Stars and symbols in box (right of each plot) indicate statistical significance to WT ESCs, calculated by two-way ANOVA with Dunnett's multiple comparison test (for A – C) or multiple unpaired t-tests (for D): ns (non-significant, $p>0.05$), *$p<0.05$, **$p<0.01$, ***$p<0.001$, ****$p<0.0001$. The
*Figure 2 continued on next page*

*Figure 2 continued*

same WT ESCs data is presented in **A** – **D** for comparison. (**E**) Cell movement, presented as mean squared displacement per minute (MSD) for WT ESCs co-cultured with Wnt3a- or iWnt3a-beads, or Wnt/β-catenin pathway KO ESCs co-cultured with Wnt3a-beads. $n \geq 36$ cells, pooled from $\geq 3$ independent experiments, were tracked for 3 hr after attachment. Box represents median and quartiles, error bars are 10–90 percentile, dots are data outside 10–90 range. ns indicates non-significant ($p > 0.05$) differences calculated by one-way ANOVA with Dunnett's multiple comparison test. Numeric data used in the figure can be found in the Source Data file.

The online version of this article includes the following source data for figure 2:

**Source data 1.** Numeric data used in *Figure 2*.

100% of cells retaining beads in Lrp5KO #2 and 80.4% cells retaining beads in Lrp6KO #2 (*Figure 4J*).

Altogether, our results show that WT ESCs recruit and retain Wnt3a-beads at the membrane, where they are positioned near the centre of the cell with limited movement. However, WT ESCs do not recruit iWnt3a-beads efficiently and form unstable interactions with them, positioning them at the periphery of the cells, where they are also dropped at higher rates. While perturbation of the pathway components has a limited effect in recruited Wnt3a-bead movement and Wnt3a-bead retention, KO ESCs exhibit a significantly impaired ability to position Wnt3a-beads close to the centre of the cell.

## AMPA/Kainate-type receptors are required for dynamic interactions with localised Wnt3a in ESCs

A crosstalk between Lrp6 and the activity of ionotropic AMPA/Kainate-type glutamate receptors (iGluRs) regulates the reactivity of ESC cytonemes to Wnt3a-beads (*Junyent et al., 2021*; *Junyent et al., 2020*). Here, we aimed to assess whether iGluR activity might also regulate the dynamics of Wnt3a-stem cell interaction.

ESCs express the AMPA subunits *Gria3* and *4*, and the Kainate subunits *Grik1*, *2*, *3*, and *5* (*Junyent et al., 2021*; *Junyent et al., 2020*). Previous publications have shown that GriA3/4 combinations can form active AMPA receptors that channel $Ca^{2+}$ (*Brorson et al., 2004*; *Cull-Candy et al., 2006*; *Liu and Zukin, 2007*), while Kainate receptor formation requires the presence of either GriK1 or GriK3 (*Lerma and Marques, 2013*).

To investigate the role of AMPA and Kainate iGluRs in Wnt3a-ESC interaction, we pharmacologically inhibited AMPA/Kainate iGluR activity by adding cyanquixaline (CNQX) to the media, which reduces the generation of localised, Wnt3a-mediated $Ca^{2+}$ transients in the ESC cytonemes and impairs Wnt3a-bead recruitment (*Junyent et al., 2020*). To dissect the specific role of each type of receptor family, we generated ESC lines with a double KO for GriA3/4 (GriA3/4dKO, clones #1 and #2) and GriK1/3 (GriK1/3dKO, clones #1 and #2, *Figure 5—figure supplement 1*).

We initially explored the capacity of these cells to activate the Wnt/β-catenin pathway in response to exogenous Wnt3a (200 ng/mL solubilised Wnt3a or a high density of Wnt3a-beads, 6.25 μg/cm², *Figure 5A–C*) after 24 hr. In these conditions, cells were cultured as a monolayer, and therefore Wnt3a-cell contact is cytoneme independent (*Figure 5C*; *Junyent et al., 2021*; *Junyent et al., 2020*). We analysed the relative expression of *Axin2* by qPCR (for GriA3/4dKO and GriK1/3dKO) or the production of eGFP in response to 7xTCF-eGFP activation by FACS (for CNQX-treated cells). In all conditions, Wnt3a addition led to activation of the Wnt/β-catenin pathway, to levels comparable to WT (*Figure 5A and B*).

Next, we examined whether perturbation of AMPA/Kainate receptors by KO or pharmacological inhibition affected the capacity of ESCs to contact and recruit Wnt3a-beads. At the single-cell level, both GriK1/3dKO and GriA3/4dKO ESCs generate thin dynamic cytonemes of similar number and length to WT ESCs (*Figure 6A and B*, *Figure 1—figure supplement 1*, *Videos 7* and *8*). These results corroborate previous observations in CNQX-treated WT ESCs (*Junyent et al., 2020*), suggesting that AMPA/Kainate iGluRs do not play a role in cytoneme formation.

At the population level, GriK1/3dKO ESCs, or WT ESCs + CNQX, generate cytonemes at a rate similar to WT ESCs (approx. 44% of cells with cytonemes after 4 hr, *Figure 6C,E and F*), while the percentage of cells with cytonemes in GriA3/4dKO ESCs is equal to or higher than control cells (~67% after 4 hr for clone #2, *Figure 6D and F*). However, Wnt3a-bead accumulation is affected in Kainate-receptor–deficient cells: GriK1/3dKO ESCs show lower Wnt3a-bead accumulation than WT

ESCs, with a maximum mean of 49.5% cells with beads over the 12 hr (*Figure 6G*). This phenotype is even more pronounced in WT ESC treated with CNQX, reaching only a mean of 39.9% of cells with Wnt3a-beads after 12 hr (*Figure 6I and J*).

GriA3/4dKO ESCs show clonal variability, with clone #1 exhibiting reduced Wnt3a-bead uptake, and cells from clone #2 recruiting Wnt3a-beads at a rate similar to WT cells (*Figure 6H and J*). This is accompanied by a significant increase in motility, with a median cell MSD of ~1.4 $\mu m^2$/min for GriA3/4dKO, compared to ~1.1 $\mu m^2$/min for WT ESCs (*Figure 6K*).

Next, we explored the dynamics of Wnt3a-bead movement after initial recruitment by AMPA/Kainate receptor-deficient cells. Using high temporal resolution live imaging (as described in *Figure 4B*), we observed that both GriK1/3dKO and GriA3/4dKO ESCs rapidly relocate the Wnt3a-bead from the tip of the cytoneme to the cell soma, but stabilise the position of the bead, on average, significantly further from the cell centre than WT ESCs (~0.62 of cell radius between 30 and 120 min, *Figure 7A and B*, expanded in *Figure 7—figure supplement 1*). These cells show a slight increase in Wnt3a-bead movement, in a clone-dependent, genotype-independent manner (*Figure 7D*). In CNQX-treated ESCs, where both AMPA and Kainate iGluRs are inhibited, the Wnt3a-beads are positioned far from the cell centre (*Figure 7C*), and with restricted movement (*Figure 7D*).

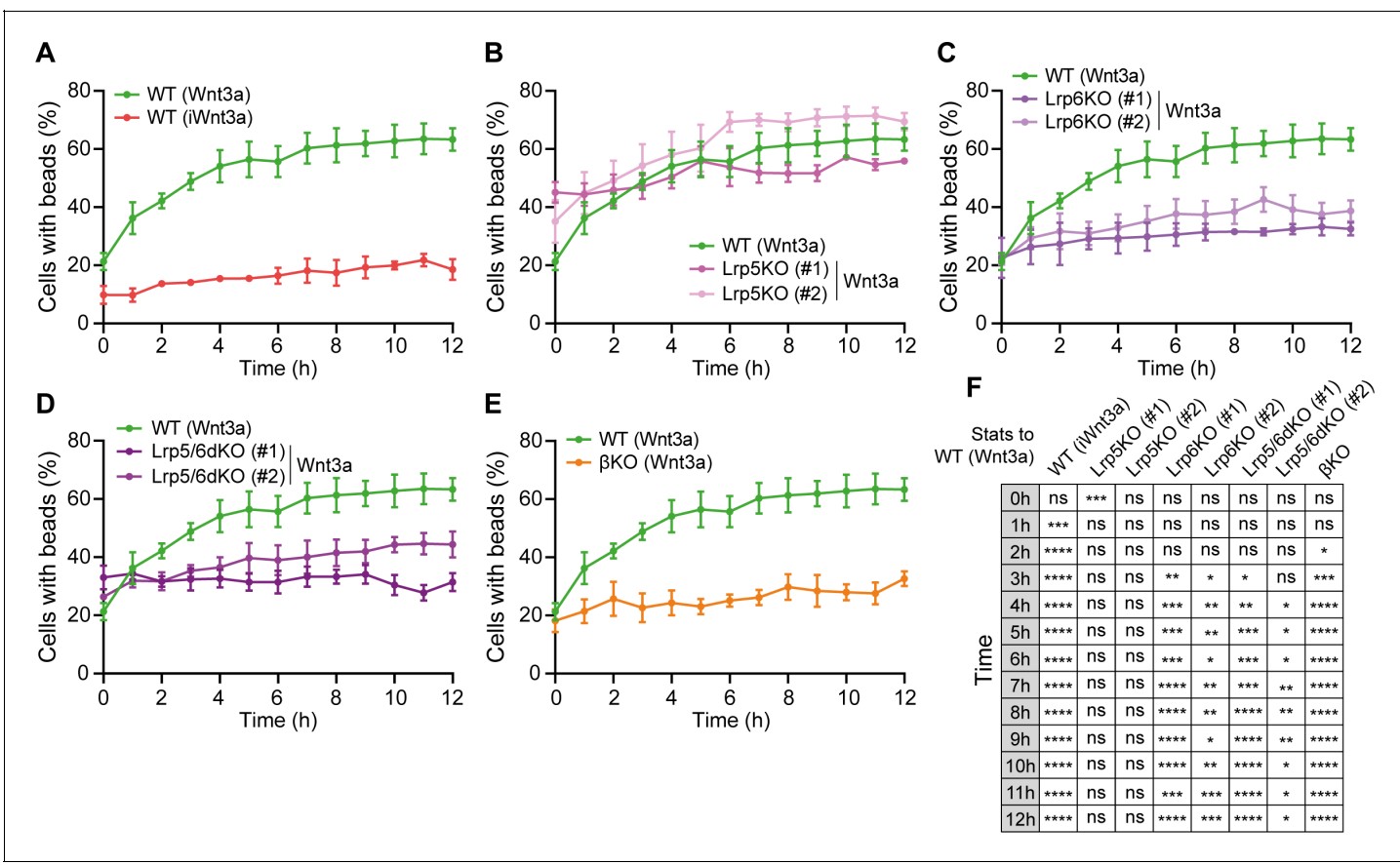

**Figure 3.** Lrp6 and β-catenin are required for Wnt3a-bead accumulation. (A – E) Percentage of WT (**A**), Lrp5KO (#1 and #2) (**B**), Lrp6KO (#1 and #2) (**C**), Lrp5/6dKO (#1 and #2) (**D**) and βKO (**E**) ESCs contacting Wnt3a-beads or iWnt3a-beads (indicated in brackets) at 0–12 hr after seeding. Line and dots are mean, error bars are SEM. N ≥ 36 cells, n ≥ 3 experiments. The same WT ESCs + Wnt3a-beads data is presented in (A – E) for comparison. (**F**) Statistical analysis of (A – E), calculated for all conditions against WT ESCs + Wnt3 a-beads. Stars and symbols indicate statistical significance calculated by two-way ANOVA with Dunnett's multiple comparison tests: ns (non-significant, p>0.05), *p<0.05, **p<0.01, ***p<0.001, ****p<0.0001. Numeric data used in the figure can be found in the Source Data file.

The online version of this article includes the following source data for figure 3:

**Source data 1.** Numeric data used in *Figure 3*.

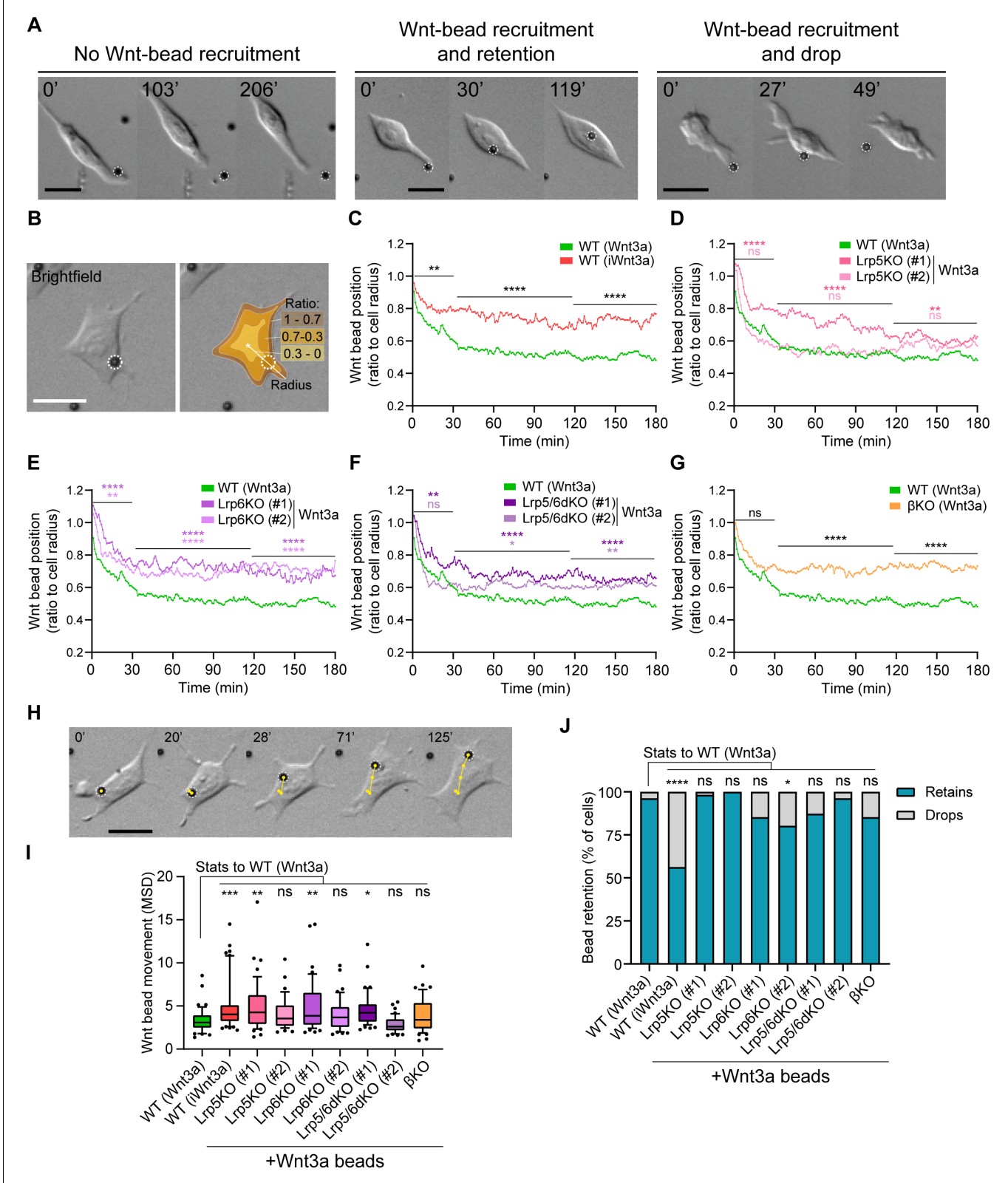

**Figure 4.** Wnt co-receptors and β-catenin orchestrate the interaction between ESCs and Wnt3a-beads at the membrane. (**A**) Representative brightfield frames of time-course live imaging sequences displaying an ESC contacting a Wnt3a-bead without recruitment (*left*), contact with recruitment (*middle*) and a Wnt3a-bead recruitment with subsequent dropping (*right*). Scale bars are 20 μm. Wnt3a-beads are highlighted by white dashed circles. (**B**) Representative brightfield image of an ESC contacting a Wnt3a-bead (*left*). Schematic representation of the cell 'periphery' (1–0.7 proportion of radius,

*Figure 4 continued on next page*

*Figure 4 continued*

brown), 0.7–0.3 proportion of radius (orange) or 'centre' of the cell (0.3–0 proportion of radius, yellow), and of the radius used to calculate the bead position (white line) (*right*). Scale bars are 20 μm. Wnt3a-bead is highlighted by white dashed circles. (C – G) Wnt3a-bead position in the cell, measured every minute for 3 hr (180 min) after initial cytoneme-mediated recruitment, and presented as ratio to cell radius. Lines are mean of n ≥ 38 cells/condition, pooled approximately evenly from ≥ 3 independent experiments. Single-cell tracks are presented in *Figure 4—figure supplement 1*. Stars or symbols are statistical significance, calculated from the average Wnt3a-bead position per cell in three representative time windows (0–30 min, 31–120 min, and 121–180 min) against the position of Wnt3a-beads in WT ESCs, by two-way ANOVA with Dunnett's (D – F) or Šídák's (C and G) multiple comparison test: ns (non-significant, p>0.05), *p<0.05, **p<0.01, ****p<0.0001. The same WT ESCs + Wnt3a-beads data is presented in C – G for comparison. (H) Representative brightfield frames of a time-course live imaging, showing an ESC contacting a Wnt3a-bead (highlighted with white dashed circle). Yellow track indicates Wnt3a-bead movement on the cell over time. Time is in min. Scale-bar is 20 μm. (I) Wnt3a-bead movement on the cell, presented as MSD. Bead movement was measured every minute for 3 hr (180 min) after cytoneme-mediated recruitment in WT ESCs contacting Wnt3a- or iWnt3a-beads, and in Wnt pathway KO ESCs contacting Wnt3a-beads. Box indicates median and quartiles, error bars indicate 10–90 percentile, dots are data outside 10–90 percentile range. n ≥ 38 cells pooled approximately evenly from ≥ 3 independent experiments. Stars indicate statistical significance against WT ESCs + Wnt3a-beads, calculated by one-way ANOVA with Dunnett's multiple comparison test: ns (non-significant, p>0.05), *p<0.05, **p<0.01, ***p<0.001. (J) Percentage of cells that drop a Wnt3a-bead or iWnt3a-bead within 3 hr of initial recruitment with a cytoneme (grey) or retain it for longer than 3 hr (blue). n ≥ 43 cells/condition. Stars indicate statistical significance calculated by multiple Fisher's exact tests: ns (non-significant, p>0.05), *p<0.05, ****p<0.0001. Numeric data used in the figure can be found in the Source Data file.

The online version of this article includes the following source data and figure supplement(s) for figure 4:

**Source data 1.** Numeric data used in *Figure 4*.
**Figure supplement 1.** Details of Wnt3a-bead position in WT or Wnt/β-catenin pathway KO ESCs membrane.
**Figure supplement 1—source data 1.** Numeric data used in *Figure 4—figure supplement 1*.

Collectively, our data indicates that KO or inhibition of AMPA/Kainate-type receptors, much like KO of Wnt co-receptor Lrp6 or pathway effector β-catenin, impairs Wnt3a-bead uptake and alters the dynamics of Wnt3a-ESC interaction at the cell membrane. In all conditions, importantly, once Wnt3a-cell contact has been achieved, it is often sustained until division, enabling downstream analysis of the interaction (*Figure 7E*).

A localised Wnt3a signal can induce spindle orientation and asymmetric cell division (ACD) of single ESCs (*Habib et al., 2013*). We next investigated WT, Wnt/β-catenin pathway and AMPA/Kainate iGluRs KO ESCs throughout the process of Wnt3a-mediated ACD.

## Wnt/β-catenin pathway components regulate the orientation of the mitotic spindle

The Wnt3a-bead orients the spindle of dividing ESCs (*Habib et al., 2013*). Spindle orientation is an important factor in the process of ACD in several biological systems (*Bertrand, 2016*; *Goldstein et al., 2006*; *Huang and Niehrs, 2014*; *Kaur et al., 2020*; *Sugioka et al., 2011*; *Walston et al., 2004*). Spindle orientation is established and can be determined as early as the ana-phase (*Kiyomitsu and Cheeseman, 2013*). Therefore, we measured the angle (α) between the minor axis of mitosis and the position of the bead, relative to the centre of division, at ana-phase (*Figure 8A and B*). In this way, 90° represents a completely Wnt3a-bead orientated division (*Figure 8B*).

WT ESCs divide predominantly oriented towards the Wnt3a-bead, with a most common angle of 85–90° (*Figure 8C*). This Wnt3a-biased distribution is significantly different to a rando-mised (Wnt3a-independent) distribution (Chi-squared test, p<0.001), in which division angle counts are assumed to be equally distributed between 0°-30°, 30°-60°, and 60°-90°. Indeed, when presented with iWnt3a-beads, a rando-mised distribution of angles is observed (Chi-squared test, p>0.05), with an increased

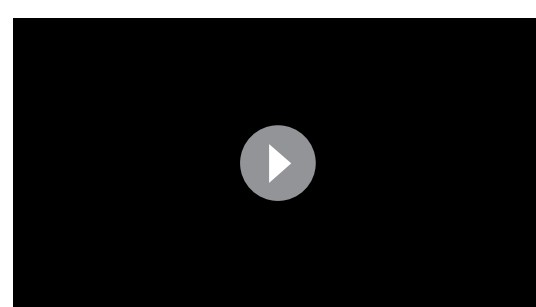

**Video 6.** Tracking of ESCs and Wnt3a-beads. Time-course live-cell brightfield images of a WT or Lrp6KO ESC interacting with a Wnt3a-bead. Cell centre and Wnt3a-bead were tracked for 180 min, and the tracks are displayed. Scale bars, 20 μm.
https://elifesciences.org/articles/59791#video6

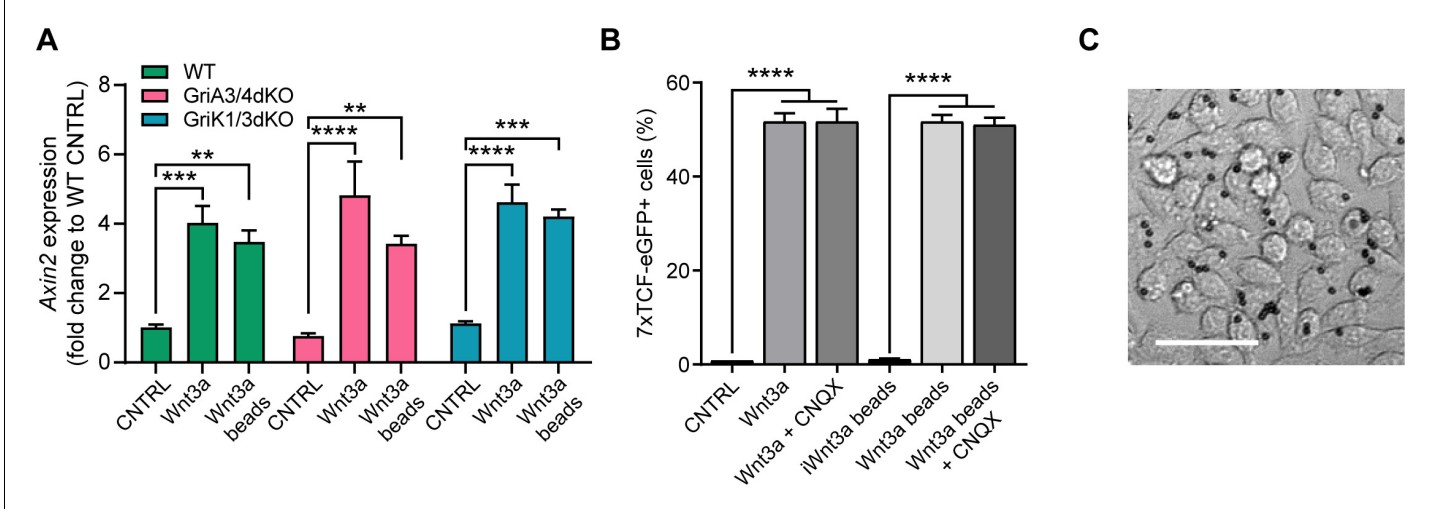

**Figure 5.** iGluR inhibition or KO does not affect cytoneme-independent Wnt3a responsiveness. (**A**) Relative expression of *Axin2* mRNA in a monolayer of WT, GriA3/4dKO or GriK1/3dKO ESCs treated with control media (CNTRL), solubilised Wnt3a (200 ng/mL) or a high density of Wnt3a-beads for 24 hr. Bars are mean of n = 3, error bars are SEM. Stars indicate statistical significance, calculated by two-way ANOVA with Tukey's multiple comparison test: **p<0.01, ***p<0.001, ****p<0.0001. (**B**) Percentage of 7xTCF-eGFP+ cells in control- or CNQX-treated cells upon addition of control solution (CNTRL), solubilised Wnt3a (200 ng/mL), Wnt3a-beads or iWnt3a-beads at coating concentrations. Bars are mean of n = 3, error bars are SEM. Stars indicate statistical significance calculated by one-way ANOVA with Šídák's multiple comparison test: ****p<0.0001. (**C**) Representative brightfield image of a monolayer of cells treated with Wnt3a-beads. Scale bar is 50 µm. Numeric data used in the figure can be found in the Source Data file.

The online version of this article includes the following source data and figure supplement(s) for figure 5:

**Source data 1.** Numeric data used in *Figure 5*.
**Figure supplement 1.** Genotype of GriA3/4dKO and GriK1/3dKO ESCs.

proportion of cells dividing with iWnt3a-beads between 0° and 5° (*Figure 8D and L*), where the contact area is maximised and the free energy is minimised (*Koltover et al., 1999*).

Single ESCs that lack Lrp5, Lrp6, or β-catenin exhibit spindle misorientation (*Figure 8E–K*), with significantly different angle distributions than WT ESCs + Wnt3a-beads, as determined by two-sample Kolmogorov-Smirnov (K-S) statistical analysis (*Figure 8L*). In these ESC lines, most cells divide at angles ranging from 0° to 60° for Lrp5KO, Lrp6KO, and Lrp5/6dKO ESCs (*Figure 8E–J*), or with randomised angle distributions (Chi squared, p>0.05) for βKO ESCs (*Figure 8K*).

In conclusion, disruption of the Wnt co-receptors or the downstream component β-catenin impairs the spindle orientation towards a source of localised Wnt3a during division. While localised Wnt3a controls spindle orientation, it also controls cell fate, producing a self-renewing Wnt3a-proximal ESC and a Wnt3a-distal cell prone to differentiation (*Habib et al., 2013*). Next, we studied whether the individual KOs affect cell fate determination during ACD.

## Lrp6, but not Lrp5, is critical for asymmetric cell fate determination

ACD is the partitioning or enrichment of cellular components in one of the two daughter cells deriving from mitosis, leading to differences in their cell fate. Wnt3a-beads induce ACD in single ESCs (*Habib et al., 2013*). Using this system, quantification of the expression of pluripotency markers Nanog, Rex1 and Sox2 (*Figure 9A–D*) allows the comparison of the cell fate of the daughter cells: a higher expression in the Wnt3a-contacting cell indicates maintenance of a naïve ESC, whereas a lower expression in the Wnt3a-distal cell indicates a shift towards a more differentiated state.

To analyse ACD in WT or Wnt/β-catenin pathway KO ESCs, we seeded single cells at low density in the presence of Wnt3a- or iWnt3a-beads for 6 hr prior to fixation. Then we analysed divided cell pairs where a bead is in clear contact with only one of the two daughter cells. This ensured that only cell pairs with an asymmetric presentation of the Wnt3a signal were analysed (*Figure 9A*). We distinguished three categories: a 'Proximal' doublet consisting of a cell contacting a Wnt3a-bead (Wnt3a-proximal) with higher pluripotency marker signal, and a Wnt3a-distal cell with reduced pluripotency marker expression. Conversely, a doublet where the farthest cell expresses more pluripotency

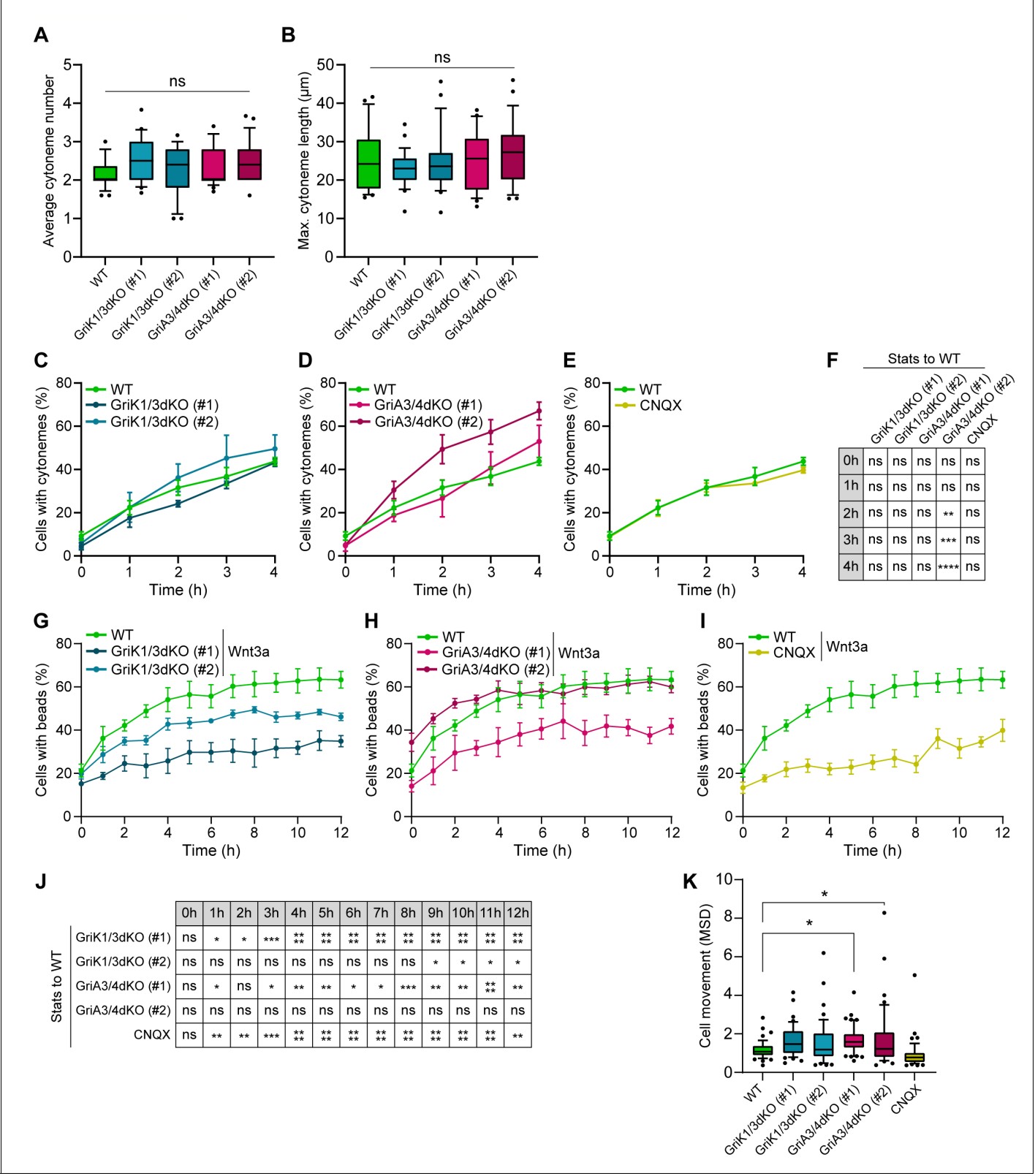

**Figure 6.** iGluRs control Wnt3a-bead recruitment without affecting the generation and characteristics of the cytonemes. (A – B) Average number of cytonemes/cell (A) and maximum cytoneme length/cell (B) in WT, GriA3/4dKO, GriK1/3dKO and CNQX-treated ESCs. Boxes are median and quartiles, error-bars are 10–90 percentiles, dots are data outside range. n ≥ 25 cells/condition (A and B). 'ns' are non-significant differences calculated against WT ESCs by one-way ANOVA with Dunnett's multiple comparison tests. (C – E) Percentage of WT, GriK1/3dKO (clone #1 and #2) (C), GriA3/4dKO (#1 and

*Figure 6 continued*

#2) (D) and CNQX-treated ESCs (E) with cytonemes at 0–4 hr after seeding. Line and dots are mean, error bars are SEM. N ≥ 38 cells, n ≥ 3 experiments. WT ESC data from *Figure 2A-D* presented for comparison. (F) Statistical analysis of (C – E). Stars and symbols in box indicate statistical significance of each condition to WT ESCs, calculated by two-way ANOVA with Dunnett's multiple comparison test (for C and D) or multiple unpaired t-tests (for E): ns (non-significant, $p>0.05$), **$p<0.01$, ***$p<0.001$, ****$p<0.0001$. (G – I) Percentage of WT, GriK1/3dKO (#1 and #2) (G), GriA3/4dKO (#1 and #2) (H) and CNQX-treated ESCs (I) contacting Wnt3a-beads at 0 to 12 hr after seeding. Line and dots are mean, error bars are SEM. N ≥ 36 cells, n ≥ 3 experiments. WT ESCs + Wnt3a-beads data from *Figure 3A-E* presented for comparison. (J) Statistical analysis of (G – I). Stars and symbols indicate statistical significance of each condition to WT ESCs, calculated by two-way ANOVA with Dunnett's multiple comparison tests: ns (non-significant, $p>0.05$), *$p<0.05$, **$p<0.01$, ***$p<0.001$, ****$p<0.0001$. (K) Cell movement, presented as MSD for WT, GriK1/3dKO, GriA3/4dKO and CNQX-treated ESCs. n ≥ 44 cells, pooled from ≥ 3 independent experiments, were tracked for 3 hr after attachment. Box represents median and quartiles, error bars are 10–90 percentile, dots are data outside 10–90 range. Stars indicate statistical differences calculated by one-way ANOVA with Dunnett's multiple comparison test: *$p<0.05$. Numeric data used in the figure can be found in the Source Data file.

The online version of this article includes the following source data and figure supplement(s) for figure 6:

**Source data 1.** Numeric data used in *Figure 6*.
**Figure supplement 1.** Representative images of cytonemes in GriK1/3dKO and GriA3/4dKO ESCs.

marker was named a 'Distal' doublet. An even distribution, with no significant bias to either cell, was named 'Distributed'. Of these, only the 'Proximal' condition represents a Wnt3a-mediated ACD.

When contacting a Wnt3a-bead, most WT ESC doublets have 'Proximal' marker expression and have therefore undergone Wnt3a-mediated ACD (*Figure 9B–D*). On the other hand, the majority of WT ESCs dividing with iWnt3a-beads are 'Distributed', with a similar partitioning of pluripotency markers between daughter cells. While Lrp5KO ESCs have a reduced ability to precisely align the spindle towards a Wnt3a signal (*Figure 8E and F*), their capacity for Wnt3a-mediated ACD is retained (*Figure 9B–D*). ESCs lacking Lrp6 or β-catenin (Lrp6KO, Lrp5/6dKO, βKO), however, show impaired ACD (*Figure 9B–D*), with cell pairs predominantly having 'Distributed' expression of Nanog, Rex1 and Sox2 between daughter cells (*Figure 9B–D*).

To validate the roles of Lrp6 and Lrp5, we transiently transfected Lrp5/6dKO ESCs with a plasmid encoding GFP and the *Lrp6* sequence and sorted them into GFP⁺ (Lrp5KO, Lrp6 overexpression, *OE*) and GFP⁻ (Lrp5/6dKO) populations (*Figure 10A and B*). Lrp5/6dKO cells overexpressing Lrp6 (GFP⁺) could not efficiently align the spindle towards the Wnt3a-bead during mitosis (*Figure 10C*). However, Wnt3a-oriented ACD was rescued in these cells, as shown by an increased rate of 'Proximal' doublets, quantified for Sox2 and Nanog (*Figure 10D–F*). Altogether, these results reiterate differences between Lrp5 and Lrp6 in the regulation of spindle orientation and ACD.

Considering all evidence, we have shown that the co-receptor Lrp6 and β-catenin play co-dependent roles in the positioning of localised Wnt3a signals at the membrane of ESCs, as well as in regulating spindle orientation and ensuring asymmetric partitioning of cell fate between the daughter cells. KO of these components in ESCs results in severely impaired activation of the Wnt/β-catenin pathway, as well as impaired recruitment of Wnt3a-beads. Lrp5KO ESCs retain higher capacity than Lrp6KO ESCs to activate the Wnt/β-catenin pathway in response to solubilised Wnt3a, to recruit Wnt3a-beads and to induce ACD, despite a compromised spindle orientation towards the Wnt3a-bead. Importantly, regardless of clonal variability observed in the initial interaction stages between the cells and the Wnt3a signals, the downstream phenotypes observed are strongly retained when the cells divide.

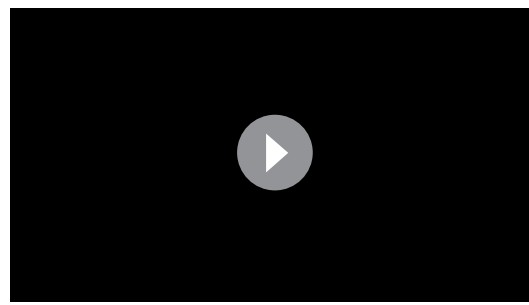

**Video 7.** GriA3/4dKO ESC cytonemes. High-quality images of the dynamic cytonemes of GriA3/4dKO ESCs (clones #1 and #2). Cells were stained and imaged as described for *Video 1*. Images are presented as inverted greyscale. Yellow boxes are magnifications, contrast-enhanced for clarity. Time is min and s. Scale bar, 10 μm.
https://elifesciences.org/articles/59791#video7

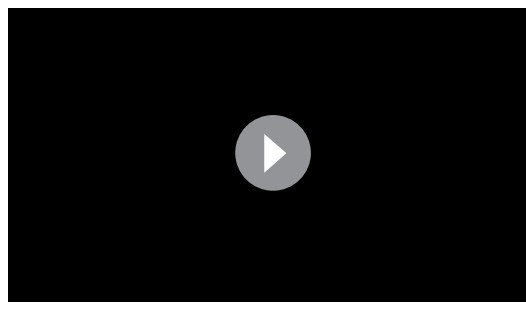

**Video 8.** GriK1/3dKO ESC cytonemes. High-quality images of the dynamic cytonemes of GriK1/3dKO ESCs (clone #1 and #2). Cells were stained and imaged as described for *Video 1*. Images are presented as inverted greyscale. Yellow boxes are magnifications, contrast-enhanced for clarity. Time is min and s. Scale bar, 10 μm.

https://elifesciences.org/articles/59791#video8

## AMPA-type iGluRs participate in controlling Wnt3a-oriented ACD

Our data has shown that AMPA/Kainate receptor activity is required for Wnt3a signal recruitment and interaction dynamics in ESCs, suggesting a pervasive crosstalk with the Wnt/β-catenin pathway activity. Next, we explored whether such effects span beyond early Wnt3a-stem cell interaction in modulating the process of division.

Wnt3a-beads fail to orient the spindle of AMPA/Kainate receptor-defective ESCs. GriK1/3dKO, GriA3/4dKO and WT ESCs treated with CNQX show misoriented spindle distributions (*Figure 11A*). In Kainate receptor KO cells, spindle orientation angles accumulate between 0° and 60°, while GriA3/4dKO ESCs or CNQX-treated cells present broader, randomised spindle angle distributions (Chi-squared test, $p > 0.05$). All AMPA/Kainate-type iGluR-defective cells are significantly different to the Wnt3a-biased spindle orientation observed in WT ESCs (K-S statistical tests, *Figure 11A*).

Post-division doublet analysis revealed that, while GriK1/3dKO ESCs largely exhibit Wnt3a-oriented ACD, GriA3/4dKO or CNQX treatment reduces the proportion of 'Proximal' doublets dividing cells (*Figure 11C–E*).

To confirm these results we treated WT ESCs with NBQX, a pharmacological inhibitor specific to AMPA receptors (*Goldstein and Litwin, 1993*). When compared to control ESCs treated with DMSO, NBQX-treated cells also fail to orientate the spindle towards a Wnt3a-bead (*Figure 11B*) or undergo Wnt3a-oriented ACD (*Figure 11F–H*).

Our data indicates that AMPA/Kainate-type iGluR activity has important roles throughout the interaction between ESCs and localised sources of Wnt3a. GriK1/3dKO impairs Wnt3a-bead recruitment without affecting the number or length of the cytonemes, or the rate of cytoneme generation in the population. AMPA-type receptors (involving GriA3/4) play lesser roles in ensuring Wnt3a-bead recruitment. Yet, both AMPA and Kainate iGluRs are important to correctly position the bead at the cell membrane, and to align the mitotic spindle towards the Wnt3a signal. Nevertheless, precise spindle orientation is also not required for Wnt3a-oriented ACD in these cells: Kainate receptors are dispensable for ACD but lack of GriA3/4 leads to reduced asymmetric cell fate marker segregation.

Together with our analysis of Wnt/β-catenin pathway KO ESCs, our results suggests that a dynamic crosstalk between AMPA/Kainate receptor signalling and Lrp6 is not only involved in signal recognition and response, as previously shown, but is important to regulate the spatial organisation and specification of dividing ESCs by Wnt3a.

## Discussion

Cell fate and spatial positioning occur in concert to achieve proper tissue organisation. In the stem cell niche, responsive stem cells receive Wnts as a cue for self-renewal (*Clevers et al., 2014*; *Garcin and Habib, 2017*). In a spatial context, Wnt reception has been shown to induce cell polarisation and ACD, in an evolutionarily conserved manner (*Goldstein et al., 2006*; *Huang and Niehrs, 2014*; *Mizumoto and Sawa, 2007*; *Ouspenskaia et al., 2016*; *Sugioka et al., 2011*; *Walston et al., 2004*).

Understanding initial ligand recruitment to stem cell membranes may benefit regenerative medicine and the targeting of Wnt-related diseases. However, the study of dynamic, localised Wnt reception by single stem cells is impeded by difficulties in visualising secreted Wnts. The Wnt3a-bead (*Habib et al., 2013*; *Lowndes et al., 2017*) allows a temporal analysis spanning the entire duration of the stem cell—Wnt3a-bead interaction: from cell morpho-motility prior to contact and initial bead

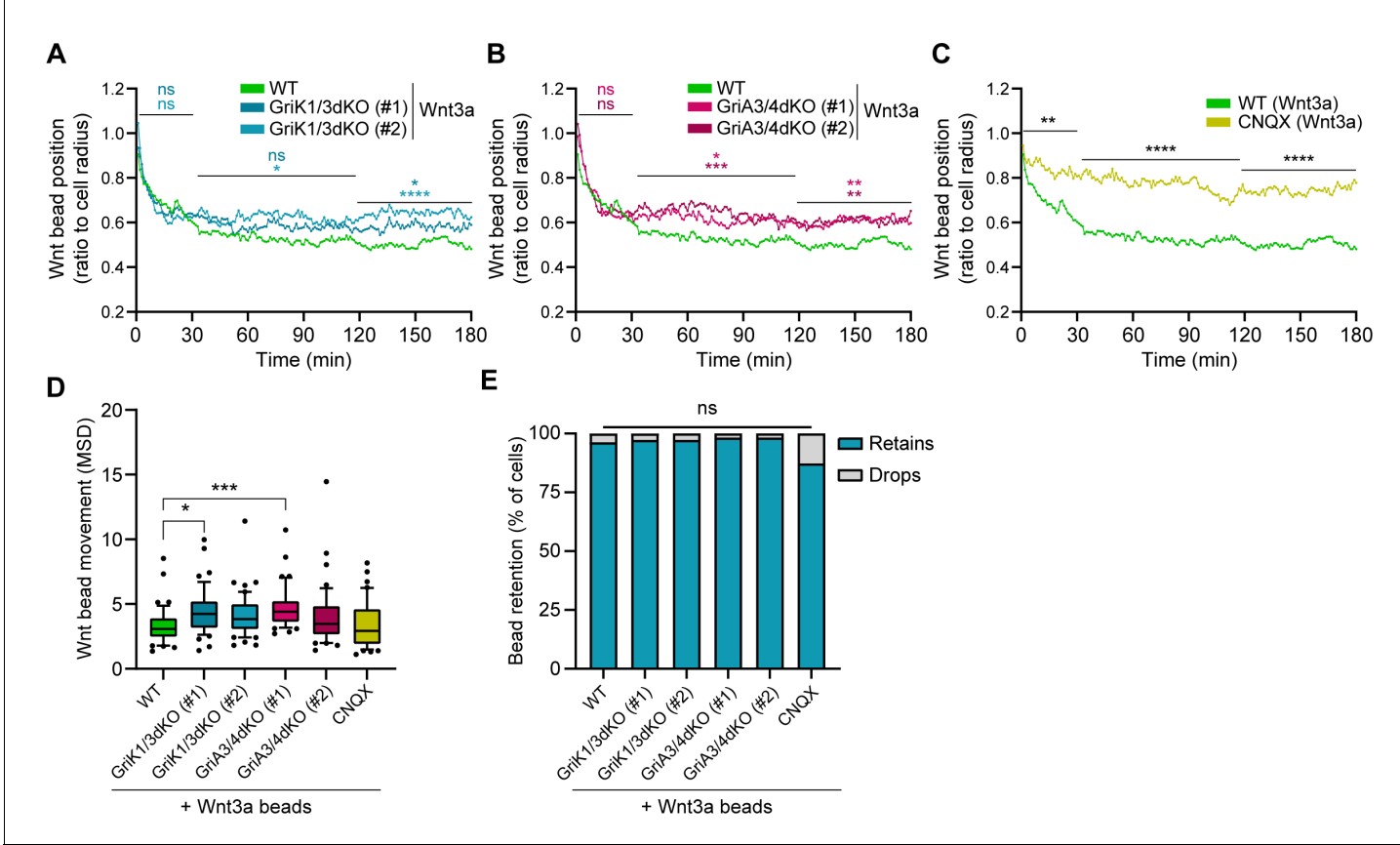

**Figure 7.** GriK1/3dKO, GriA3/4dKO or CNQX-treated cells retain the beads further from the cell centre. (A – C) Wnt3a-bead position in the cell surface, measured every minute for 3 hr (180 min) after initial cytoneme-mediated recruitment, and presented as proportion of cell radius. Lines are mean of n ≥ 45 cells/condition, pooled approximately evenly from ≥3 independent experiments. Single-cell tracks are presented in *Figure 7—figure supplement 1*. Stars or symbols are statistical significance, calculated from the average Wnt3a-bead position per cell in three representative time windows (0–30 min, 31–120 min, and 121–180 min) against the position of Wnt3a-beads in WT ESCs, by two-way ANOVA with Dunnett's multiple comparison test: ns (non-significant, p>0.05), *p<0.05, **p<0.01, ***p<0.001, ****p<0.0001. WT ESCs + Wnt3a-beads data from *Figure 4C-G* presented for comparison. (D) Wnt3a-bead movement on the cell, presented as MSD. Bead movement was measured every minute for 3 hr (180 min) after cytoneme-mediated recruitment in WT GriK1/3dKO, GriA3/4dKO and CNQX-treated ESCs contacting Wnt3a-beads. Box indicates median and quartiles, error bars indicate 10–90 percentile, dots are data outside 10–90 percentile range. n ≥ 45 cells pooled approximately evenly from ≥ 3 independent experiments. Stars indicate statistical significance against WT ESCs + Wnt3a-beads, calculated by one-way ANOVA with Dunnett's multiple comparison test: ns (non-significant, p>0.05), *p<0.05, ***p<0.001. (E) Percentage of cells that drop a Wnt3a-bead within 3 hr of initial recruitment with a cytoneme (grey) or retain it for longer than 3 hr (blue). n ≥ 37 cells/condition. ns is non-significant differences calculated by multiple Fisher's exact tests. Numeric data used in the figure can be found in the Source Data file.

The online version of this article includes the following source data and figure supplement(s) for figure 7:

Source data 1. Numeric data used in *Figure 7*.

Figure supplement 1. Details of Wnt3a-bead position in WT, GriK1/3dKO, GriA3/4dKO, and CNQX-treated ESCs membrane.

Figure supplement 1—source data 1. Numeric data used in *Figure 7—figure supplement 1*.

recruitment by cytonemes, to the dynamic interaction at the cell membrane, ultimately leading to effects on spindle orientation and ACD.

AMPA/Kainate-type iGluR activity, and its crosstalk with the Wnt/β-catenin pathway, controls cytoneme-mediated recruitment of Wnt signals in mouse ESCs (*Junyent et al., 2020*). Additionally, components of the Wnt pathway regulate the length and number of cytonemes (*Junyent et al., 2020*). Conversely, we show in this study that knockout of AMPA or Kainate receptors does not affect cytoneme formation.

The Wnt co-receptors Lrp5 and Lrp6 have mostly been studied in the context of β-catenin-dependent signalling, where they have overlapping functions (*He et al., 2004*; *Houston and Wylie, 2002*;

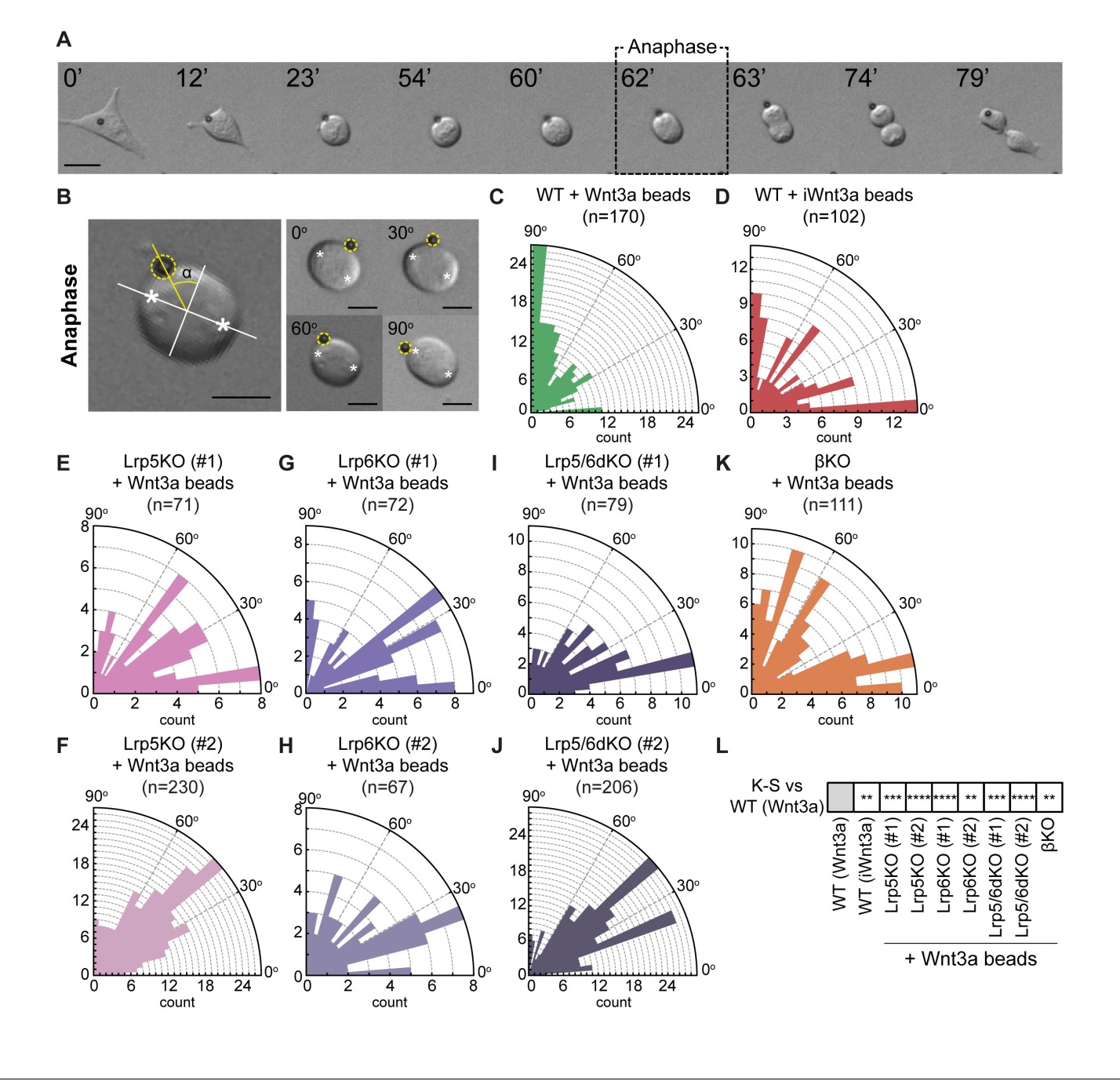

**Figure 8.** Components of the Wnt/β-catenin pathway regulate the orientation of the mitotic spindle of ESCs. (**A**) Representative brightfield frames of a time course live-cell imaging, showing an ESC contacting a Wnt3a-bead and undergoing Wnt3a-oriented division. Wnt3a-bead is black sphere. Dashed black box indicates the 'Anaphase' timepoint used for spindle orientation analysis. Scale bar is 10 μm. (**B**) (*Left*) Representative image of the method used for spindle angle measurement relative to Wnt3a-bead position. Asterisks indicate cell poles used as reference for the measurement. Orthogonal white lines depict major and minor axis, yellow line points to Wnt3a-bead and shows the angle measured (α). Wnt3a-bead is highlighted with yellow dashed circle. (*Right*) Representative images of different ESCs dividing with Wnt3a-beads at angles between 0 and 90° (indicated in the figure). Scale bars are 10 μm. (**C – K**) Rose plots depicting the distribution of spindle angles in WT ESCs dividing with Wnt3a-beads (**C**), WT ESCs dividing with iWnt3a-beads (**D**) or Lrp5KO ESCs (**E and F**), Lrp6KO ESCs (**G and H**), Lrp5/6dKO ESCs (**I and J**), and βKO ESCs (**K**) dividing with Wnt3a-beads. n (number of cells analysed) is depicted in the figure for each condition. (**L**) Statistical comparison of the distribution of spindle angles in all conditions to the one observed in WT ESCs + Wnt3a beads. Stars indicate statistical significance calculated by multiple Kolmogorov-Smirnov tests: \*\*p<0.01, \*\*\*p<0.001, \*\*\*\*p<0.0001. Numeric data used in the figure can be found in the Source Data file.

*Figure 8 continued on next page*

*Figure 8 continued*

The online version of this article includes the following source data for figure 8:

**Source data 1.** Numeric data used in *Figure 8*.

*Kelly et al., 2004*; *Tamai et al., 2000*; *Wehrli et al., 2000*), albeit to different levels of efficiency (*Holmen et al., 2002*; *MacDonald et al., 2011*). Here, we demonstrate differing roles between them: endogenous expression of Lrp6 alone in ESCs (in Lrp5KO) results in an increased rate of cytoneme formation over WT ESCs, while cells lacking Lrp6 (Lrp6KO and Lrp5/6dKO) have a reduced rate of cytoneme formation, suggesting a distinct role for Lrp6 in cytoskeleton regulation.

Both Lrp5 and Lrp6 contain cytoplasmic domains that can bind to regulators of actin filaments (*Brown et al., 1998*; *Krause et al., 2003*; *Lian et al., 2016*; *Zhu et al., 2012*), however this region of Lrp5 has just 37% identity to Lrp6, compared to 69% overall sequence identity between them (*MacDonald et al., 2011*). We speculate that such differences might contribute to their differential roles in cytoneme formation/elongation: Lrp5 may compete with Lrp6 for the interaction of Wnt-pathway components or cytoskeletal regulators. Hence, improved Lrp6-cytoskeleton interactions may facilitate the increased rate of cytoneme formation and initial Wnt3a-bead recruitment observed in Lrp5KO ESCs.

WT ESCs use cytonemes to efficiently recruit Wnt3a-beads, and subsequently position them near the centre of the cell. iWnt3a-beads cannot be recruited with the same efficiency. Additionally, iWnt3a-beads are maintained nearer the periphery of the cell, with shorter and less stable interactions. AMPA/Kainate receptor activity is involved throughout the Wnt3a-interaction. AMPA or Kainate receptor KO ESCs, similarly to those lacking Lrp6 or β-catenin, are able to recruit Wnt3a-beads via cytonemes, but over 12 hr are inefficient in their population-wide accumulation of Wnt3a-signal compared to WT or Lrp5KO ESCs.

Beyond the initial interaction between cell and Wnt3a-ligand, a role for AMPA/Kainate receptors lies in controlling the dynamics of Wnt3a on the membrane. Unlike WT ESCs, neither AMPA nor Kainate receptor KO cells are able to relocate the Wnt3a-bead from the cytoneme-periphery and retain it near the centre of the cell. Pharmacological inhibition of both AMPA and Kainate receptors together by CNQX resulted in a similar phenotype with an enhanced effect. The KO of Wnt-pathway components Lrp5, Lrp6 or β-catenin also reduces the ability to position a Wnt3a-bead on the membrane near the centre of the cell.

Following ligand-receptor binding, Wnt-pathway component polarisation (*Capelluto et al., 2002*) and oligomerisation (*Cong et al., 2004*) are suggested to play roles in the enhancement or modulation of Wnt signalling (*Bilic et al., 2007*). Here, the spatial stabilisation and colocalisation of upstream and downstream pathway components at a 'signalling hub' may be critical for efficient signal transmission. In fact, we previously demonstrated colocalisation of Lrp6 and AMPA/Kainate receptors during ESC interaction with a Wnt3a-bead (*Junyent et al., 2021*).

Membrane receptors, including glutamate receptors (*Yang et al., 2006*), have been demonstrated to flow from membrane protrusions towards the centre of the cell. This motion can be promoted by directed movement of the actin cytoskeleton (*Hanley, 2014*; *Yu et al., 2010*) and its efficiency can be governed by the activation of the pathway (*Hartman et al., 2009*). Areas of relative membrane concavity, such as the cell centre away from the periphery, have also been shown to enhance membrane receptor interactions with intracellular components (*Rangamani et al., 2013*). This process may be required for more efficient signalling (*Schmick and Bastiaens, 2014*). Therefore, the control of the Wnt3a-bead positioning in WT ESCs may facilitate efficient protein interactions required for signal transduction, enhancing downstream pathway activation.

After Wnt3a-bead recruitment and upon progression to mitosis, WT ESCs orientate the spindle towards the source of localised Wnt3a at anaphase. Perturbation of the AMPA/Kainate receptors or Wnt/β-catenin pathway components resulted in significant impairment of the alignment of the spindle to localised Wnt3a, suggesting that both pathways function in concert during this process.

Interestingly, AMPA receptors are involved in Wnt3a-mediated ACD. While Kainate receptor KO had no significant effect, both AMPA receptor KO and AMPA receptor–specific inhibition by NBQX significantly compromised Wnt3a-mediated ACD. Similarly, Wnt3a-mediated ACD is impaired when ESCs lack Lrp6 or β-catenin. Lrp5KO ESCs, however, much like with Kainate receptor KO, remain

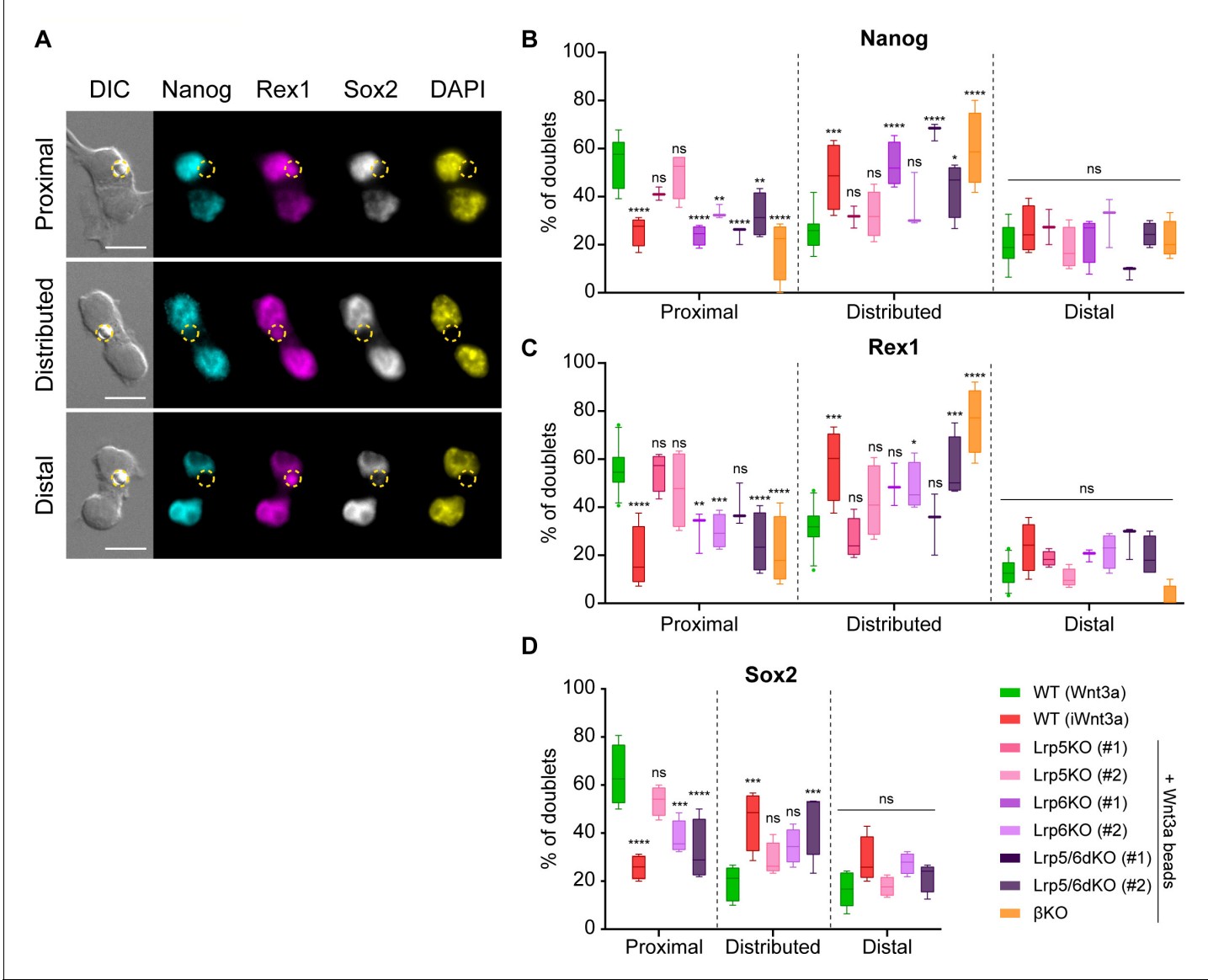

**Figure 9.** β-catenin and Lrp6 are required for Wnt3a-mediated ACD. (**A**) Representative images of ESC doublets after dividing in contact with a Wnt3a-bead, labelled with antibodies against Nanog (cyan, AF488), Rex1 (magenta, AF555) and Sox2 (greyscale, AF647), or with 4',6-diamidino-2-phenylindole (DAPI, yellow). Images are maximum intensity projections of a selected Z-stack. DIC is differential interference contrast. Panels display doublets with higher marker expression in the Wnt3a-bead contacting cell (*top*, 'Proximal'), with equal levels of expression between both cells (*middle*, 'Distributed') or with higher marker expression in the cell not contacting the Wnt3a-bead (*bottom*, 'Distal'). The Wnt3a-beads are highlighted by yellow dashed circles. Scale-bars are 20 μm. (**B – C**) Percentage of doublets of WT or Wnt/β-catenin KO ESCs contacting Wnt3a-beads or iWnt3a-beads in the 'Proximal', 'Distributed' or 'Distal' categories for Nanog (**B**), Rex1 (**C**) or Sox2 (**D**). n ≥ 3 experiments, with N ≥ 18 doublets/n (Nanog, Rex1) or N ≥ 28 doublets/n (Sox2). For D. only clone #2 of Lrp5KO, Lrp6KO, and Lrp5/6dKO ESCs were analysed. Box indicates median and quartiles, error bars indicate 10–90 percentile. Dots are data outside range. Stars and symbols indicate statistical significance for each condition against WT ESCs + Wnt3a beads, calculated by two-way ANOVA with Dunnett's multiple comparison test: ns (non-significant, $p > 0.05$), *$p < 0.05$, **$p < 0.01$, ***$p < 0.001$, ****$p < 0.0001$. Numeric data used in the figure can be found in the Source Data file.

The online version of this article includes the following source data for figure 9:

**Source data 1.** Numeric data used in *Figure 9*.

able to asymmetrically segregate pluripotency markers to the proximal cell, despite the lack of precise alignment of the spindle towards a Wnt3a-bead at anaphase. Therefore, a robust spindle orientation towards a Wnt3a-bead is not required for asymmetric fate outcome. Our data further demonstrates a differential requirement for Lrp5 and Lrp6 in ESCs, beyond the rate of cytoneme

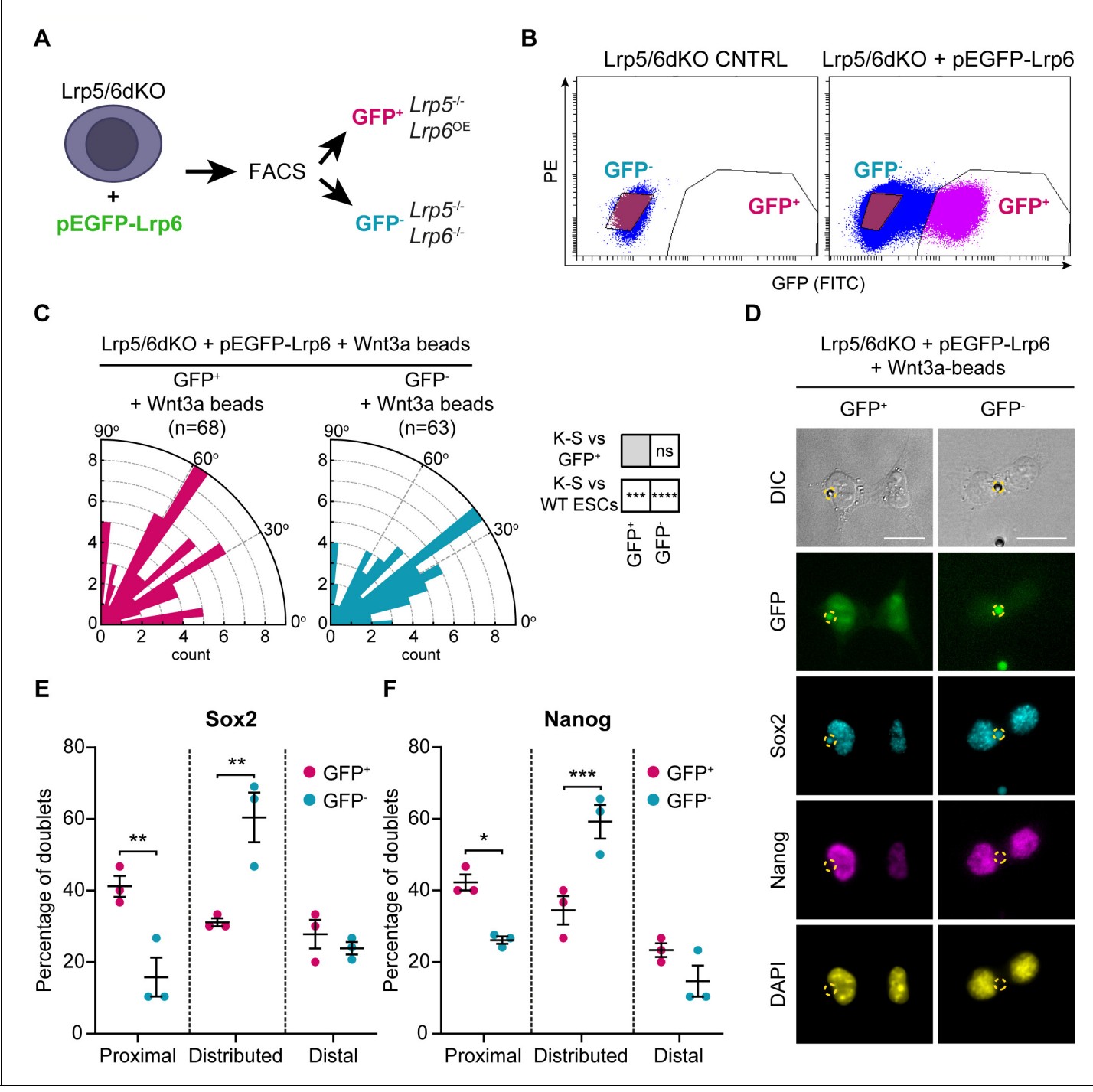

**Figure 10.** Lrp6 overexpression in Lrp5/6dKO ESCs rescues ACD. (A) Schematic of the experimental pipeline. Lrp5/6dKO (#2) ESCs were transfected with pEGFP-Lrp6 plasmids and sorted by FACS to *Lrp6*-expressing (GFP⁺) and *Lrp6* non-expressing (GFP⁻) populations before experimentation. (B) Representative dot-plot of GFP expression levels in Lrp5/6dKO control (non-transfected) ESCs or Lrp5/6dKO ESCs transfected with pEGFP-Lrp6. GFP intensity was detected using a FITC filter. Y axis is PE. GFP⁻ (blue) and GFP⁺ (magenta) cells were sorted only in the transfected condition. (C) Rose plots depicting the distribution of spindle angle orientations in GFP⁺ or GFP⁻ ESCs dividing in contact with a Wnt3a-bead. Stars and symbols in boxes (*right*) indicate statistical significance calculated by multiple Kolmogorov-Smirnov tests against GFP⁺ ESCs (*top*) or WT ESCs dividing with a Wnt3a-bead (*bottom*): ns (non-significant, $p>0.05$), ***$p<0.001$, ****$p<0.0001$. n (number of cells) is depicted in the figure. (D) Representative images of GFP⁺ and GFP⁻ ESCs after dividing with a Wnt3a-bead, stained with antibodies against Sox2 (cyan) and Nanog (magenta), or with DAPI (yellow). Wnt3a-beads are highlighted by yellow dashed circle. Scale bars are 20 μm. (E and F) Percentage of doublets in the 'Proximal', 'Distributed' or 'Distal' categories (described in *Figure 9A*) in GFP⁺ and GFP⁻ ESCs stained for Sox2 (E) and Nanog (F). Mean of n = 3 independent experiments, N ≥ 29 doublets

*Figure 10 continued*

analysed per n, error bars are SEM. Stars indicate statistical significance calculated by two-way ANOVA with Šídák's multiple comparison tests: *$p<0.05$, **$p<0.01$, ***$p<0.001$. Numeric data used in the figure can be found in the Source Data file.

The online version of this article includes the following source data for figure 10:

**Source data 1.** Numeric data used in *Figure 10*.

formation, in the regulation of cell fate. Interestingly, the *Drosophila* neuroblast can tolerate certain degrees of spindle misorientation due to disruption in internal polarity cues during early stages of cell division. These cells can overcome initial misorientation and retain the prospective cell fate of the daughter cells (*Knoblich et al., 1995*; *Peng et al., 2000*; *Schober et al., 1999*).

In summary, response to localised Wnt3a reception and the downstream Wnt3a-mediated ACD is not orchestrated by the Wnt/β-catenin pathway alone, but also by crosstalk with AMPA/Kainate-type glutamate receptor activity.

Glutamate receptor activity is an evolutionarily conserved pathway that links homologous proteins in prokaryotes (e.g. GluR0 *Chen et al., 1999*) and plants (GLRs *Forde and Roberts, 2014*; *Ortiz-Ramírez et al., 2017*) with metazoans (iGluRs). In mammals, they are well known for their function in excitatory synaptic signalling (*Niciu et al., 2012*). However, there is a more ancient role for glutamate signalling that pervades throughout single- and multi-cellular organisms alike: chemotaxis.

From non-neuronal mammalian cells (*Gupta and Chattopadhyay, 2009*), to plant reproductive cells (*Ortiz-Ramírez et al., 2017*) and bacteria (*Brumley et al., 2019*), sensitivity to glutamate or related molecules can provide a spatial signal that influences cellular organisation. In *E. coli*, cytoplasmic proteins are segregated to the cellular poles by interaction and colocalisation with glutamate-sensitive chemoreceptors (*Maddock and Shapiro, 1993*). This therefore presents a glutamate-receptor–mediated mechanism for the spatial orientation of intracellular components in response to external signals. Further, glutamate acts as a nutrient (*Hottes et al., 2004*) and chemoattractant (*Felzenberg et al., 1996*) to the bacterium *Caulobacter crescentus*. This bacterium undergoes ACD to replicate its 'stalked' form, and to generate a motile 'swarmer' cell. Crucially, in *C. crescentus*, chemoreceptors are synthesised in the pre-divisional cell (*Gomes and Shapiro, 1984*), before being asymmetrically polarised at division to the pole of the cell along with other protein components of the swarmer cell fate (*Alley et al., 1992*). It remains to be determined whether the glutamate-sensing machinery is specifically required for the ACD of *C. crescentus* and other bacteria. However, these findings above suggest an evolutionarily conserved link between the sensing of external chemoattractants, such as glutamate, and the polarisation and compartmentalisation of cell fate components.

As this ancient function for glutamate receptor activity is present in ESCs during Wnt3a-mediated ACD, it may prove to be a mechanism that influences stem cell biology, cell fate and tissue organisation throughout mammalian adult stem cells and metazoans.

## Materials and methods

### Cell culture

Wild-type W4 (129S6/SvEvTac) mouse ESCs (RRID:CVCL_Y634) or ESCs knock-out for Lrp5, Lrp6 or Lrp5/6dKO (generated as described in *Junyent et al., 2020*) were maintained in ESC basal media containing Advanced DMEM/F12 (cat. num. 12634028, ThermoFisher), 10% ESC-qualified foetal bovine serum (FBS, cat. num. ES-009-B, Millipore), 1% penicillin-streptomycin (cat. num. P4333, Sigma), 2 mM GlutaMax (cat. num. 35050061, ThermoFisher), 50 µM β-mercaptoethanol (cat. num. 21985–023, ThermoFisher) and 1000 U/mL recombinant Leukaemia Inhibitory Factor (LIF; cat. num. 130-095-775, Miltenyi), supplemented with with 2i: MEK inhibitor PD0325901 (1 µM, cat. num. 130-104-170, Miltenyi) and GSK3 inhibitor CHIR99021 (3 µM, cat. num. 130-104-172, Miltenyi). Heterozygous β-catenin deficient (β^fl/-) ESCs, with an inducible flox β-catenin, and stably carrying a Cre-ER-T2 cassette under the control of a chicken β-actin promoter (generated in a W4 129S6/SvEvTac background by *Raggioli et al., 2014*) were cultured in the same conditions. Media was changed daily, and cells were grown at low density until formation of mid-sized colonies before passaging (every 3–4 days). To passage ESCs, colonies were rinsed with PBS, treated with 0.25% trypsin-EDTA (T-E, cat.

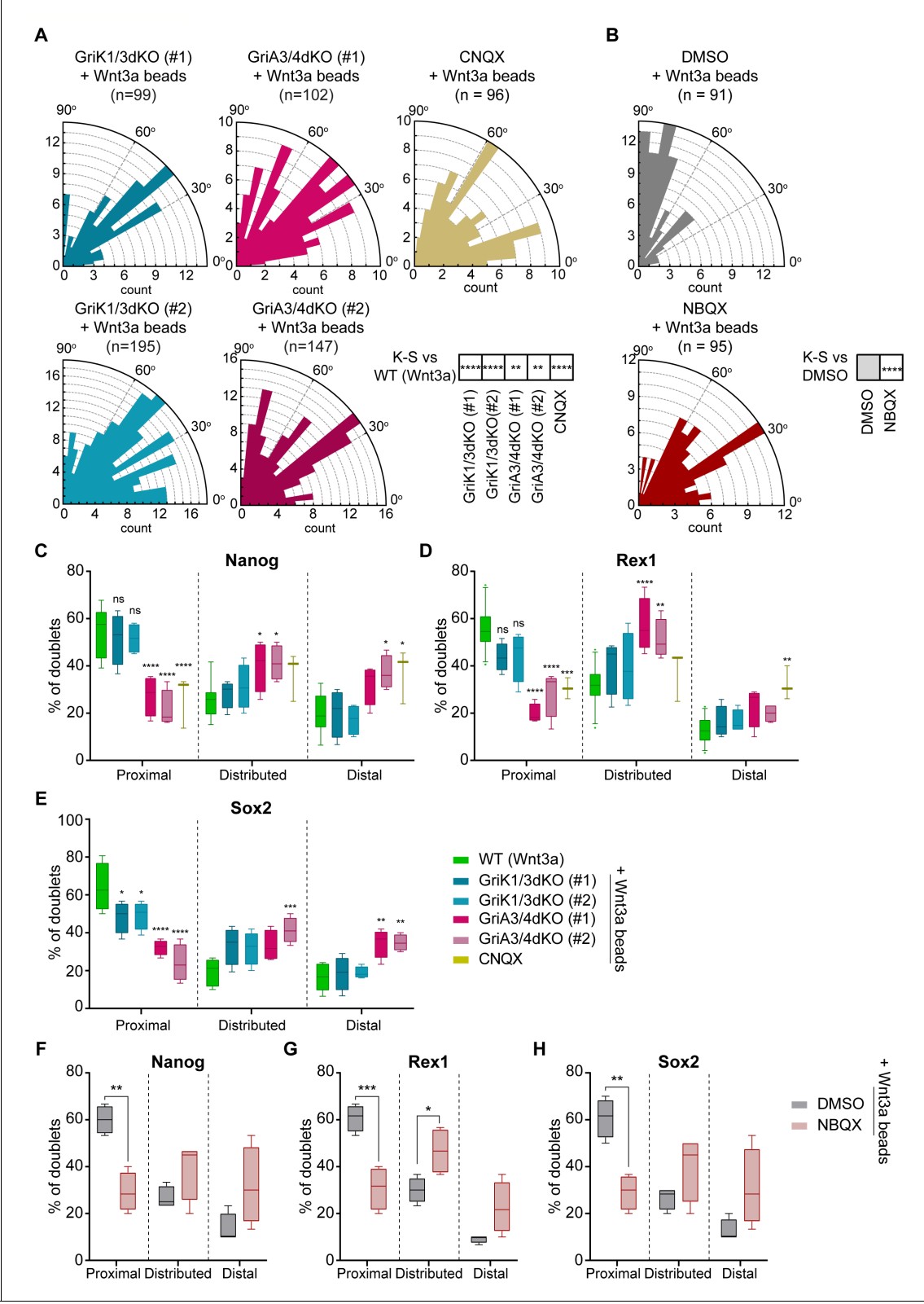

**Figure 11.** AMPA-type iGluR receptors are important for Wnt3a-oriented ACD. (**A and B**) Rose plots depicting the distribution of spindle orientations in GriK1/3dKO, GriA3/4dKO or CNQX-treated ESCs (**A**) or DMSO- or NBQX-treated ESCs (**B**) dividing with a Wnt3a-bead. n (number of cells analysed) is stated in the figures. Stars in boxes indicate statistical significance calculated by multiple Kolmogorov-Smirnov test against WT ESCs + Wnt3a-beads (in **A**) or DMSO-treated ESCs + Wnt3a-beads (in **B**): **p<0.01, ****p<0.0001. (**C – E**) Percentage of doublets in the 'Proximal', 'Distributed' or 'Distal'

*Figure 11 continued on next page*

*Figure 11 continued*

categories (described in *Figure 9A*) for WT, GriK1/3dKO, GriA3/4dKO, or CNQX-treated ESCs that have divided with Wnt3a-beads. Quantification for Nanog (C), Rex1 (D), or Sox2 (E). n ≥ 3 experiments, with N ≥ 18 doublets/n (Nanog, Rex1) or N ≥ 30 doublets/n (Sox2). Box indicates median and quartiles, error bars indicate 10–90 percentile. Dots are data outside range. Stars and symbols indicate statistical significance calculated by two-way ANOVA with Dunnett's multiple comparison test: ns (non-significant, *p>0.05*), *p<0.05, **p<0.01, ***p<0.001, ****p<0.0001*. WT ESC data from *Figure 9B-D* presented for comparison. (F – H) Percentage of doublets in the 'Proximal', 'Distributed' or 'Distal' categories (described in *Figure 9A*) for DMSO- or NBQX-treated ESCs that have divided with a Wnt3a-bead. Quantification for Nanog (F), Rex1 (G) or Sox2 (H). n = 4 experiments, with N = 30 doublets/n. Box indicates median and quartiles, error bars indicate 10–90 percentile. Stars indicate statistical significance calculated by two-way ANOVA with Šídák's multiple comparison tests: *p<0.05, **p<0.01, ***p<0.001*. Numeric data used in the figure can be found in the Source Data file. The online version of this article includes the following source data for figure 11:

**Source data 1.** Numeric data used in *Figure 11*.

num. 25200056, ThermoFisher) for 4 min at 37°C, 5% $CO_2$, resuspended using ESC basal media and centrifuged at 1200 x *g* for 4 min. Pelleted cells were resuspended in ESC basal media, counted and seeded in a clean tissue culture-treated 6-well plate at low density (~7000 cells/well) in complete ESC media (containing LIF and 2i). In some experiments, 24 hr before use, 2i was withdrawn and replaced with soluble Wnt3a protein (200 ng/mL).

β-catenin knock-out in β$^{fl/-}$ cells was induced through the addition of 0.1 μg/mL 4-hydroxy-tamoxifen (4-HT, cat. num. H6278, Sigma) to the media every 24 hr for 72 hr. Cells treated for 3 days with 4-HT were used in the experiments (referred to as βKO in the text).

All cell lines were maintained at 37°C, 5% $CO_2$ and were routinely tested for mycoplasma infection.

## Pharmacological inhibition of iGluRs

When indicated in the text, figures, or figure legends, cells were treated with the AMPA/Kainate iGluR inhibitor cyanquixaline (CNQX, 10 μM, cat. num. C127, Sigma) or the AMPA-iGluR-specific inhibitor 2,3-dioxo-6-nitro-7-sulfamoyl-benzo[f]quinoxaline (NBQX, 40 μM, cat. num. 373, Tocris). Drugs were dissolved in ESC complete media (no 2i) to the desired concentration and were added to the cells at the beginning of the experiment. For Wnt/β-catenin pathway activation experiments, CNQX was added to the cells 1 hr before solubilised Wnt3a or Wnt3a-bead addition.

## Generation of KO ESCs by CRISPR/Cas9-mediated gene editing

Generation of KO ESCs was achieved by CRISPR/Cas9-mediated gene editing, following *Ran et al., 2013*. gRNA sequences adjacent to a protospacer adjacent motif (PAM) and targeted at an early conserved exon in *Lrp5*, *Lrp6*, *Gria3*, *Gria4*, *Grik1*, and *Grik3* were designed in Geneious 11 (Biomatters; RRID:SCR_010519). gRNAs were scored for on- and off-target effects against the whole mouse genome, and sequences with low off-target score were synthesised as oligonucleotides and cloned into a PX458 plasmid (Addgene plasmid #48138; RRID:Addgene_48138). gRNA sequences used can be found in *Figure 1—figure supplement 1* and *Figure 5—figure supplement 1*.

WT (to generate KO for *Lrp5*, *Lrp6*, *Gria3*, and *Grik3*), Lrp6KO (to generate the Lrp5/6dKO line), GriA3KO (to generate GriA3/4dKO cells), or GriK3KO (to generate GriK1/3dKO) ESCs were transfected with 1 μg PX458 plasmid containing the sequence for the correct gRNA, using JetPrime (cat. num. 114–07, Polyplus transfection) according to manufacturer instructions. After 24 hr, transfected cells were sorted (as described in the 'Fluorescence activated cell sorting' section), and single GFP⁺-cells were collected in the wells of a 96-well tissue culture-treated plate, in ESC complete media with LIF and 2i. Individual clones were grown until formation of colonies, then expanded by passaging twice. Genomic DNA was extracted using a DNeasy Blood and Tissue extraction kit (cat. num. 69506, Qiagen) and the targeted region was amplified by *Taq* polymerase PCR (cat. num. M7501, Promega). Sequence verification was achieved by TOPO-cloning (cat. num. 450159, ThermoFisher) of the *Taq*-PCR fragments, transformation into Top10 chemically competent bacteria (cat. num. C404006, ThermoFisher) and clonal selection. At least 12 TOPO-clones per mutant clone were analysed by Sanger sequencing (Source Bioscience). Clonal ESC lines with homozygous mutations that generated a frame-shift mutation inducing a premature stop codon were selected for the experiments. KO ESCs were cultured as described above.

## Preparation of Wnt3a-microbeads

Recombinant Wnt3a proteins were produced as described before (*Habib et al., 2013*; *Junyent et al., 2020*), or purchased (cat. num. 1324-WN, R&D systems). Wnt3a proteins were immobilised to 2.8 µm carboxylic acid–coated Dynabeads (cat. num. 14305D, ThermoFisher), as described before (*Habib et al., 2013*; *Junyent et al., 2020*; *Lowndes et al., 2017*). Briefly, the carboxylic acid groups on the Dynabeads were activated by a 30 min incubation with N-(3-Dimethylaminopropyl)-N′-ethylcarbodiimide (EDC, cat. num. 39391, Sigma) and N-Hydroxysuccinimide (NHS, cat. num. 56480–25G, Sigma). 50 mg/mL EDC and NHS solutions were prepared in 25 mM cold 2-(N-morpholino) ethanesulfonic acid (MES, cat. num. M3671-50G, Sigma), at pH 5. EDC/NHS incubation was performed in constant rotation at room temperature (RT). Following activation, beads were retained using a magnet and washed three times with 25 mM MES buffer (pH 5). Solubilised Wnt3a protein (500 ng) was diluted 1:5 in MES buffer (pH 5) and incubated with the beads for 1 hr in constant rotation, at RT. Beads were washed again three times with PBS (pH 7.4) before storage in media containing 10% FBS at 4°C. Inactivation of Wnt3a beads was achieved through incubation with 20 mM Dithiothreitol (DTT; Life Technologies, #P2325) for 30 min at 37°C. Following incubation with DTT, beads were washed three times in PBS before storage in media containing 10% FBS at 4°C (up to 10 days). Before use in the experiments, Wnt3a-bead activity was validated as described before (*Habib et al., 2013*; *Junyent et al., 2020*; *Lowndes et al., 2017*).

## Time-course live-cell imaging

To analyse the dynamics of ESCs and Wnt3a-bead interaction, 2500 ESCs/well mixed with 0.3 µg Wnt3a-beads/well were seeded in a tissue culture-treated, clear-bottomed, black-walled 96-well plate (cat. num. 3603, Corning) in complete media containing LIF (no 2i). Cells and beads were allowed to settle at 37°C, 5% $CO_2$ for 30 min. Then, the plate was placed in a Zeiss inverted Axio Imager epifluorescence microscope, equipped with a CoolSNAP HQ2 CCD camera. Between 20 and 30 positions containing cells and beads were selected and cells were imaged using a 10x/0.3 dry objective, for brightfield, every 1 min for 12 hr. The Zen 2 (Blue edition, Zeiss) software was used for acquisition. The resulting videos were analysed in Fiji (ImageJ; RRID:SCR_002285) as described in the next sections. Three or more independent videos were generated for each cell line or condition (detailed in the figure legends and the source data files).

To image the dynamics of the cytonemes in ESCs, 5,000 ESCs were seeded in a well of an ibi-Treat-modified µ-Slide eight well slide (cat. num. 80826, Ibidi) in ESC complete media with LIF (no 2i). Cells were incubated at 37°C, 5%$CO_2$ for 4 hr to allow cell attachment and cytoneme generation. Cells were stained with CellMask Deep Red plasma membrane stain (1x concentration, cat. num. C10046, ThermoFisher) for 15 min at 37°C, 5% $CO_2$. Cells were then rinsed once with ESC complete media and imaged immediately in a Nikon Eclipse Ti Inverted Spinning Disk confocal (equipped with a Yokogawa CSU-1 disk head and an Andor Neo sCMOS camera, and an incubation system at 37°C, 5% $CO_2$). Cells were imaged using a Plan Apo lambda 40x/0.95 dry objective every 10 s, for 10 min. The resulting videos were processed in Fiji (ImageJ) and presented in inverted greyscale.

## Image analysis: number and length of the cytonemes in ESCs

Brightfield time-course videos (at 1 min resolution) were used to measure the average number and maximum length of the cytonemes in WT, GriK1/3dKO and GriA3/4dKO ESCs. Cells from each condition that attached and generated cytonemes were selected. For each cell, the number and length of their cytonemes was measured from the cell centre, 10 times in timepoints separated by 10 min. The distance between the cell centre and the membrane was also measured in these timepoints. The average number of cytonemes, as well as the maximum length for any cytoneme was calculated, and the average cell centre to cell membrane distance was subtracted from the values. The normalised values are reported in the figure.

## Image analysis: percentage of cells with cytonemes or Wnt3a-beads over time

Brightfield videos of ESCs and Wnt3a-beads at 1 min resolution were also used to measure the percentage of cells presenting cytonemes or contacting Wnt3a-beads. Cytoneme generation was analysed every hour for the first 4 hr of each video (timepoints 0–4 hr), while Wnt3a-bead contact was

analysed for the totality of the video (timepoints 0–12 hr). Several positions were analysed for each independent video, on Fiji (ImageJ), to count the total number of cells in the frame, the number of cells with cytonemes and/or the number of cells in contact with Wnt3a-beads. Cell counts in the different positions of each video were summed to avoid position-specific differences, and the percentage values were calculated per time-point, as presented in the figures.

## Image analysis: Wnt3a-bead position and movement in the cell and cell movement

Single ESCs that recruited a Wnt3a-bead with a cytoneme and retained contact with it for $\geq$180 min were selected to measure Wnt3a-bead position and movement on the membrane. The same cells were used to measure cell movement. For each cell and each minute, the position of the cell centre, the Wnt3a-bead and the membrane were registered as X and Y coordinates using the 'Manual tracking' plugin on Fiji (ImageJ).

First, the distances between the cell centre and the Wnt3a-bead (Distance$_{cell-bead}$) or the cell centre and the membrane (Distance$_{cell-membrane}$, cell radius) were calculated for each minute as:

$$Distance_{cell-bead} = \sqrt{\left((x_{bead} - x_{cell})^2 + (y_{bead} - y_{cell})^2\right)}$$

$$Distance_{cell-membrane} = \sqrt{\left((x_{membrane} - x_{cell})^2 + (y_{membrane} - y_{cell})^2\right)}$$

Then, the ratio between the Distance$_{cell-bead}$ and the Distance$_{cell-membrane}$ (cell radius) was calculated and reported in the figure.

Second, cell movement was calculated as mean squared displacement (MSD), based on the position of the cell centre (P$_{cell}$) over the 180 min analysed, as:

$$MSD_{cell}(\tau) = \langle \Delta P_{cell}(\tau) \rangle = \left\langle [P_{cell}(t + \tau) - P_{cell}(t)]^2 \right\rangle$$

where $\tau$ is the interval of time between two positions (1 min).

Finally, the normalised position of the Wnt3a-bead relative to the position of the cell (Normalised position$_{bead}$, NP$_{bead}$) was calculated for each time point (t) as:

$$Normalised\,position_{bead}(t) = NP_{bead}(t) = (x_{bead}(t) - x_{cell}(t), y_{bead}(t) - y_{cell}(t))$$

In this way, the movement of the Wnt3a-bead was normalised to the movement of the cell. Then, the MSD of the Wnt3a-bead was calculated with $\tau = 1$ min as:

$$MSD_{bead}(\tau) = \langle \Delta NP_{bead}(\tau) \rangle = \left\langle [NP_{bead}(t + \tau) - NP_{bead}(t)]^2 \right\rangle$$

## Image analysis: percentage of bead retention/dropping

Brightfield videos of ESCs at 1 min resolution were also used to measure the percentage of cells that retain or drop the Wnt3a-bead after recruitment. A sample of cells from each condition were observed, and the time of Wnt3a-bead uptake with a cytoneme was registered (t$_{uptake}$). Then, the cell was observed for a further 180 min (t$_{uptake+180}$): if a cell lost contact with the Wnt3a-bead before t$_{uptake+180}$, the cell was qualified as 'drops'; if contact was maintained beyond t$_{uptake+180}$, the cell was qualified as 'retains'. Cells that divided within the observed period were excluded from the analysis. The percentage of cells that drop or retain Wnt3a-beads

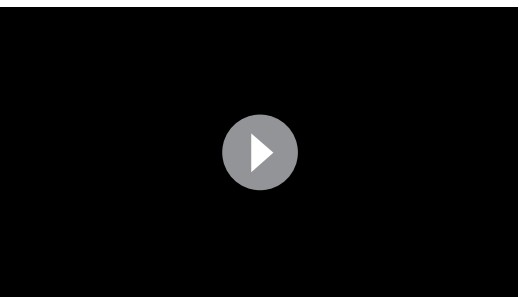

**Video 9.** Spindle orientation in dividing ESCs with Wnt3a-beads. Time-course live-cell bright-field images of a WT ESC dividing with Wnt3a-beads at ~90°, ~45° or ~0°. Spindle orientation angle was always measured at anaphase. White lines in stills indicate minor and major axis of the cell. Yellow line indicates the position of the Wnt3a-bead relative to the minor axis. Scale bars, 20 µm.
https://elifesciences.org/articles/59791#video9

over the total number of cells analysed was presented in the figures.

## Image analysis: spindle orientation

The videos at 1 min resolution were also used to measure spindle orientation. Cells dividing in contact with a Wnt3a-bead were analysed. The 'Angle' tool in Fiji (ImageJ) was used to measure the angle between the minor axis and the Wnt3a-bead position at anaphase for all conditions (*Figures 8* and *10* and *Video 9*). Rose-plots were generated with a custom Python script (included as *Source code 1*) run in Jupyter 5.0.0 using Python 2.6.

## Immunofluorescence and measurement of asymmetric cell division

A total of 3000 ESCs/well mixed with 0.3 µg Wnt3a-beads/well were seeded onto human fibronectin (10 µg/mL, cat. um. 7340101, VWR/SLS) coated 8-well slides (cat. num. 80828, Thistle Scientific) in ESC complete media containing LIF (no 2i). Slides were incubated for 6 hr at 37°C, 5% $CO_2$. After incubation, media was gently aspirated, and cells were fixed with 4% paraformaldehyde for 8 min at RT. Cells were then washed three times in staining buffer (PBS with 0.1% Bovine Serum Albumin (cat. num. A8806, Sigma), 0.05% $NaN_3$ (cat. num. S2002, Sigma) and 0.001% Tween20 (cat. num. P2287, Sigma)) for 5 min each. Slides were incubated in primary antibody diluted in staining buffer at 4°C overnight. Then, slides were carefully washed 3 times for 5 min in staining buffer and incubated with AlexaFluor (AF)-conjugated secondary antibodies diluted in staining buffer, for 1 hr at RT. Final washes were performed with staining buffer containing DAPI (cat. num. D1306, Thermo-Fisher) at 1:1000 dilution. Slides were carefully mounted with ProLong Gold antifade mounting media (cat. num. P10144, Life Technologies) and let to solidify for 24–48 hr before imaging.

Slides were imaged on a Zeiss inverted Axio Imager epifluorescence microscope using Zen 2 (Blue edition, Zeiss) software and equipped with a CoolSNAP HQ2 CCD camera. A 40x/1.3 oil immersion objective was used. Z-stacks were taken according to the optimal interval. Images were processed, deconvolved, and quantified using Volocity software (Perkin Elmer) or analysed in Fiji (ImageJ).

Quantification involved reconstruction of the cell based on 'Sum' projection and manual identification of divided cells. Cells were qualified as 'Proximal', 'Distributed' or 'Distal' by three independent researchers according to the partitioning of Nanog, Rex1 or Sox2 intensities between the Wnt3a-bead-proximal and -distal cell. Additionally, samples of doublets were allocated by semi-automated 3D intensity quantification in the Wnt3a-bead proximal and -distal cell using Volocity (Perkin Elmer; RRID:SCR_002668) or Fiji (ImageJ). In these cases, mean fluorescent intensity in the proximal and distal cells was measured for each channel and normalised to background intensity. The percentage of protein partitioning in the proximal or distal cell was used to allocate doublets as follows: 'Proximal', >55% of protein in bead-proximal cell; 'Distributed', 55% to 45% of protein in bead-proximal cell; 'Distal',<45% of protein in bead-proximal cell.

## Antibodies

The antibodies (Ab) used in this study were anti-Nanog (monoclonal rat Ab, clone eBioMLC-51; 1:500, cat. num. 14-5761-80, ThermoFisher; RRID:AB_763613), anti-Rex1 (rabbit polyclonal Ab, 14HCLC; 1:1500, cat. num. 710190, ThermoFisher; RRID:AB_2532615), anti-Rex1 (rabbit polyclonal Ab; 1:500, cat. num. ab28141, Abcam; RRID:AB_882332), anti-Sox2 (mouse monoclonal Ab, clone 245610; 1:500, cat. num. MAB2018, R&D systems; RRID:AB_358009) and AF488-, AF555- and AF647-tagged secondary antibodies (donkey; 1:1000, cat. num. A-21208, A-31572, A-31571, ThermoFisher; RRID:AB_141709, RRID:AB_162542, RRID:AB_162543).

## Fluorescence activated cell sorting

Fluorescence activated cell sorting (FACS) was employed to detect Wnt3a response in WT, KO or CNQX-treated ESCs stably expressing 7xTCF-eGFP//SV40-mCherry (*Junyent et al., 2020*). Following incubation with control media (ESC basal media), media containing solubilised Wnt3a (200 ng/mL), or addition of a high density of Wnt3a- or iWnt3a-beads (6.25 µg/cm$^2$) for 24 hr, cells were harvested using 0.25% T-E, filtered and analyzed using a FACSFortessa system (BD Biosciences). The gating strategy included gating for SSC-FSC (cells), SSC-A-SSC-W (single cells), SSC-DAPI (alive cells), SSC-PE Texas Red (mCherry$^+$ cells), and SSC-FITC (eGFP). GFP$^{-/+}$ gate was established using

CNTRL-treated and Wnt3a-treated WT ESCs. Analysis was performed using the FlowJo software (FlowJo; RRID:SCR_008520).

FACS was also used to purify GFP$^+$ and GFP$^-$ populations from an Lrp5/6dKO ESC line transfected with pEGFP-Lrp6 (*Habib et al., 2013*), or to sort single GFP$^+$ cells transfected with PX458 CRISPR/Cas9 plasmids. Cells were transfected with 1 μg pEGFP-Lrp6 or PX458 using JetPrime, 24 hr before sorting. Then, a cell suspension was prepared as described before, and cells were sorted using a FACSAriaII system (BD technologies). The gating strategy included gates for cells, single cells, alive cells and GFP (as described before), but laser intensities were adjusted as required. GFP$^{-/+}$ gates were established using control (non-transfected) ESCs and cells transfected with the plasmids. For pEGFP-Lrp6 sorting, only Lrp5/6dKO ESCs transfected with the plasmid were used for sorting GFP$^-$ and GFP$^+$ cells. Sorted cells were collected in complete ESC media containing LIF (no 2i), counted and seeded for experiments (as described before). For PX458 sorting, single cells were sorted in tissue culture-treated 96-well plates in ESC complete media (with LIF and 2i).

## Quantitative PCR

Quantitative PCR (qPCR) was used to measure *Axin2* expression in response to Wnt/β-catenin pathway stimulation. $2.0 \times 10^4$ WT or KO ESCs/well were seeded in a tissue culture-treated 12-well plate in ESC complete media supplemented with LIF (no 2i). Plates were incubated at 37°C, 5% $CO_2$ for 24 hr. Then, media was aspirated and replaced with ESC complete media (LIF, no 2i) with or without 200 ng/mL solubilised Wnt3a. Cells were returned to the incubator for 24 hr. Cell lysis, RNA extraction, and purification was performed with an RNeasy mini kit (cat. num. 74106, Qiagen), following manufacturer instructions. 500 ng RNA were retrotranscribed with a QuantiTect reverse transcription kit (cat. num. 205313, Qiagen) following manufacturer instructions. qPCR was prepared with 2x TaqMan Fast Universal PCR MasterMix (cat. num. 4366072, ThermoFisher) and TaqMan probes for *Axin2* (Mm00443610_m1, cat. num. 4331182, ThermoFisher) and GAPDH (Mm99999915_g1, cat. num. 4351370, ThermoFisher). qPCR was performed in a Bio-Rad CFX384 system (Bio-Rad). The threshold-cycle values (Ct) of *Axin2* were normalised first to Ct values of GAPDH (DCt), and then to the average DCt values of the WT CNTRL condition (DDCt) and were reported as $2^{-DDCt}$ in the figures.

## Statistical analysis

Data representation and statistical analysis were performed using Prism (GraphPad; RRID:SCR_002798), as described in the figures. ANOVA (one- or two-way) analysis, followed by Tukey's, Dunnett's or Šídák's multiple comparisons test was performed to compare data between conditions. For analysis of spindle orientation, the Kolmogorov-Smirnov statistical test was performed to compare between conditions of non-normally distributed data and Chi-square analysis was performed to compare between observed data and the expected/randomised distribution. The statistical analysis used for each assay is described in the figure legends. In all experiments, biological replicates (n) for each assay comprise of experiments performed independently from one another, and as such are from different cultured populations/passages of cells, different sessions of imaging/measurement, and independent media/reagent/cell dilution. Sample sizes (N) were selected such that the power of the statistical analysis performed was sufficient, in accordance with previous studies (*Habib et al., 2013*; *Junyent et al., 2020*). When indicated in the figure legends, the cells analysed were pooled approximately evenly from at least three independent experiments, and analysis were performed on the pooled data. Outliers were considered as individual data points significantly outside the range of possible/meaningful values for each assay and were excluded from subsequent analysis. For all figures, symbols indicate statistical significance as follows: *$p<0.05$, **$p<0.01$, ***$p<0.001$, ****$p<0.0001$. Non-significance was set as $p>0.05$. All numerical source data, as well as statistical analysis results including adjusted p-values can be consulted in the source data file for each figure.

## Acknowledgements

This work was supported by a Sir Henry Dale Fellowship (102513/Z/13/Z) to SJH. We acknowledge financial support from the Department of Health via the National Institute for Health Research (NIHR) Comprehensive Biomedical Research Centre award to the Guy's and St Thomas National

Health Service Foundation Trust in partnership with King's College London and the King's College Hospital NHS Foundation Trust. We thank Dr. Ignacio Bordeu for his help with the Python script.

## Additional information

### Funding

| Funder | Grant reference number | Author |
|---|---|---|
| Wellcome Trust | Sir Henry Dale Fellowship (102513/Z/13/Z) | Shukry J Habib |
| Royal Society | Sir Henry Dale Fellowship (102513/Z/13/Z) | Shukry J Habib |

The funders had no role in study design, data collection and interpretation, or the decision to submit the work for publication.

### Author contributions
Sergi Junyent, Joshua C Reeves, Data curation, Formal analysis, Validation, Investigation, Visualization, Methodology, Writing - original draft, Writing - review and editing; James LA Szczerkowski, Formal analysis, Validation, Visualization, Methodology; Clare L Garcin, Data curation, Formal analysis, Validation, Visualization; Tung-Jui Trieu, Data curation, Validation, Visualization; Matthew Wilson, Formal analysis; Jethro Lundie-Brown, Formal analysis, Validation, Visualization, Writing - review and editing; Shukry J Habib, Conceptualization, Data curation, Formal analysis, Supervision, Funding acquisition, Investigation, Visualization, Methodology, Writing - original draft, Project administration, Writing - review and editing

### Author ORCIDs
Sergi Junyent (iD) https://orcid.org/0000-0003-2405-5885
Joshua C Reeves (iD) https://orcid.org/0000-0001-7373-576X
Jethro Lundie-Brown (iD) https://orcid.org/0000-0003-3403-1094
Shukry J Habib (iD) https://orcid.org/0000-0003-3132-2216

### Decision letter and Author response
Decision letter https://doi.org/10.7554/eLife.59791.sa1
Author response https://doi.org/10.7554/eLife.59791.sa2

## Additional files

### Supplementary files
• Source code 1. Custom Python script used to generate Rose-plots.

• Transparent reporting form

### Data availability
All data generated or analysed during this study are included in the manuscript and supporting files.

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
