## [Decision Letter]

**Acceptance summary:**

The study by Junyent et al. provides a comprehensive analysis of the roles of the Wnt co-receptors LRP5 and LRP6, its downstream effector b-catenin as well as the ionotropic glutamate receptors (iGluRs) AMPA and Kainate receptors in the dynamic process of Wnt-ligand recruitment and retention via specialized cytonemes in embryonic stem cells and their role in oriented asymmetric cell division (ACD) using microbead-tethered Wnt ligands. Understanding the mechanism of how Wnts affect cell fate choice is of great interest in light of the well-established role of Wnt signaling in many aspects of cell biology including embryogenesis and the increasing interest in generating synthetic organs on a chip. Intriguingly, the crosstalk between Wnt and glutamate receptors in ACD in ESCs and thus the link between the sensing of external chemo-attractants and polarization and compartmentalization of cell fate components suggest that these processes are evolutionary conserved. The authors have included a tremendous amount of additional novel data including analysis of ESC lines with knock-outs of the Kainate subunits GriK1 and GriK3, the AMPA subunits GriA3 and GriA4 as well as of additional knock-out clones for LRP5, LRP6 and LRP5/6 in the revised manuscript, which support and advance their original findings.

**Decision letter after peer review:**

[Editors’ note: the authors submitted for reconsideration following the decision after peer review. What follows is the decision letter after the first round of review.]

Thank you for submitting your work entitled "Wnt-and Glutamate-receptors orchestrate stem cell dynamics and asymmetric cell division" for consideration by *eLife*. Your article has been reviewed by 3 peer reviewers, one of whom is a member of our Board of Reviewing Editors, and the evaluation has been overseen by a Reviewing Editor and a Senior Editor. The reviewers have opted to remain anonymous.

Our decision has been reached after consultation between the reviewers. Based on these discussions and the individual reviews below, we regret to inform you that your work will not be considered further for publication in *eLife*.

The study by Junyent et al. provides a comprehensive analysis of the roles of the Wnt co-receptors LRP5 and LRP6, its downstream effector b-catenin as well as ionotropic glutamate receptors (iGluRs) in the dynamic process of Wnt-ligand recruitment/retention via specialized cytonemes in embryonic stem cells and their role in oriented asymmetric cell division (ACD) using microbead-tethered Wnt ligands. Understanding the mechanism of how Wnts affect cell fate choice is of great interest in light of the well established role of Wnt signaling in many aspects of cell biology including embryogenesis and the increasing interest in generating synthetic organs on a chip. Although this manuscript describes several novel findings, the reviewers have raised the concern that some data are very similar to those published by the same group in PNAS earlier this year, and that the biological relevance of the findings is not clear. A major revision of the manuscript focusing on the novel data, a clear demonstration of the link between Wnt-regulated cytoneme formation and ACD and additional functional data would significantly improve the manuscript.

*Reviewer #1:*

The manuscript by Junyent et al. is a follow-up study of a paper by the same group recently published in PNAS. The PNAS manuscript showed that embryonic stem cells (ESCs) use specialized cytonemes to interact/pair with trophoblast stem cells (TSCs), which is the basis for spatial organization and specification of embryonic tissues. They further described cytoneme selectivity to self-renewing Wnt signals and clarified the initial mechanism, which is Lrp6-dependent and requires activity of AMPA/Kinate receptors (ionotropic glutamate receptors, iGluRs).

The current manuscript goes beyond this to study the dynamics of cytoneme formation and retention of the Wnt ligand on the membrane, which mediates asymmetric cell division. They dissect the individual roles of Lrp5, Lrp6 and iGluR activity in these processes. Here they reveal a novel function of Lrp5 in limiting the rate of cytoneme formation and also in ligand retention. Moreover, they show that accurate spindle orientation is not required for cell fate decisions. Although Lrp5KO results in a misoriented spindle, as long as the Wnt is retained on one side of the cell and can activate Wnt/b-catenin signalling, ACD occurs. Furthermore, as far as I'm aware, the cross-talk between iGluR and Wnt signalling in Wnt ligand retention and oriented ACD, and thus cell fate decision has not been studied before.

In general, the experiments seem well performed and controlled.

The authors have dissected the role of the Wnt pathway components and iGluRs in cytoneme formation and also in ACD/cell fate. What is not clear is whether the 2 processes are interlinked. Can the authors establish this relationship?

*Reviewer #2:*

In their manuscript entitled "Wnt-and Glutamate-receptors orchestrate stem cell dynamics and asymmetric cell division" Junyent et al. report a live cell imaging based approach to delineate the contribution of Lrp5, Lrp6, β catenin and iGluR in Wnt mediated asymmetric cell division. Experiments reported in this manuscript appear technically sound and experiments in Figures 4 and 5 provide interesting aspects about Wnt mediated asymmetric cell division in stem cells. However, I have some concerns that substantially diminish my enthusiasm for this paper and feel that the advance in our understanding of asymmetric cell division in stem cells provided by this manuscript is quite limited:

1. My main point of criticism is that it is not made clear which aspects and conclusions of this manuscript are substantially novel over the PNAS paper published earlier this year by the same group (Junyent et al., 2020 PNAS). Specifically, data in Figures 1 and 2 report data very similar to Figure 4 of the PNAS manuscript. The individual contributions and differential roles of Lrp5 and Lrp6, the involvement of β-catenin and the effects of CNQX treatment have also been reported in the PNAS paper. This manuscript under review feels like an annex to the PNAS paper rather than a stand-alone story. The authors should take care not to generate the appearance of publishing duplicate findings and focus only on novel data and novel key conclusions.

2. Figure 5 shows that the previously reported deficiency in establishing asymmetric cell division in Lrp5, Lrp6 and β-catenin KOs and CNQX treated cells has direct consequences on the distribution of Nanog between daughter cells. One aspect that is surprising tough is the asymmetric distribution of Rex1. Rex1 is to my knowledge a non-functional component of the pluripotency circuit (unlike Nanog) and therefore a rather surprising choice for experimentally addressing its asymmetric distribution. I am wondering what could be the developmental rational for such an asymmetric distribution of what seems purely a naïve pluripotency marker, rather than a constituent of the naïve network. Rex1 staining appears largely cytoplasmic in the proximal and distributed panels in Figure 5A, which is surprising for a transcription factor. This should be clarified/commented on. Have the authors assessed the distribution of more functional components of the naive GRN?

3. Experiments on KO ESCs without performing rescues (by restoration of the endogenous locus or by expression of the missing gene from a transgene) carry the risk of studying artefacts or clonal effects. Regardless of whether KO ESCs show the expected effect or not, these experiments are essential. Alternatively, multiple KOs, generated using different gRNAs can be used to ascertain causative relationships between genotype and phenotype. Along these lines, WT cells used in this study (W4) are not the proper control for β catenin inducible KO ESCs. Parental cells before 4OHT treatment must be used as control.

*Reviewer #3:*

The manuscript 'Wnt- and Glutamate-receptors orchestrate stem cell dynamics and asymmetric cell division' by Junyent et al. presents a comprehensive analysis on the role of WNT co-receptors Lrp5 and Lrp6, WNT signalling mediator b-catenin, and ionotropic glutamate receptors (iGluRs) in mouse embryonic stem cells (ESCs). In particular, the authors made use of a previously established approach based on microbead-tethered WNTs to expose ESCs to localised WNT3a to study the dynamics of cytoneme formation, WNT3a bead recruitment and stabilisation, and WNT-oriented asymmetric cell division in live-imaging experiments.

Generally, the manuscript is well written and the data is appropriately presented. The manuscript describes a number of observations on the kinetics of WNT-bead interactions in in LRP5, LRP6, LRP5/6dKO and b-catenin KO ESCs as well as ESCs treated with iGluR inhibitor CNQX. However, I am not sure about how conclusive the data is in terms of biological relevance as the only functional assay is the analysis of expression of two pluripotency markers after asymmetric cell division. For example, WT ESCs undergo asymmetric cell division (ACD) predominantly oriented towards the WNT3a beads. In contrast, spindle orientation in ESCs without LRP5, b-catenin or iGluRs activity appears randomized. However, this randomized spindle orientation does not have an impact on the asymmetric distribution of pluripotency markers in LRP5 KO ESCs but in ESCs without iGluRs activity.

This manuscript is a follow up study to the group's recent paper published in PNAS (Junyent et al. 2020). My concern is that some data seems to be redundant to the results presented in the PNAS paper. This manuscript describes WNT3a responsiveness, number and length of cytonemes, WNT3A-bead interactions in LRP5, LRP6, LRP5/6dKO and b-catenin KO ESC lines. As far as I can see, this was already shown in the PNAS paper. Similar videos submitted for this manuscript showing protrusions in each of the KO cell lines were published alongside the PNAS paper. Having said that, I appreciate that this manuscript provides additional insight and a more comprehensive analysis.

Another concern is technical: The numbers in the source data do not always match the numbers presented in the plots or mentioned in the text. Generally, these are minor differences (see specific comments), but nevertheless need to be carefully revised. Also, the source data for Figure 5 includes data on Dsh2 KO cells, which is not included in this manuscript.

1. The manuscript implies that KO ESC lines were generated for this study (e.g. p1 line 18: 'Through live imaging and gene editing, we show…'; p.5 line 6-: '…we knocked-out (KO) the co-receptors…as previously described.'). However, as far as I can see, these cell lines were generated for the PNAS study (Junyent et al. 2020) and re-used for this study, which needs to be more clearly phrased.

2. Figure 1B,C: Were the betaKO cells treated for 3 days with 4-HT before WNT3a addition?

3. Figure 1C: Numbers in source file do not exactly match some numbers in plot or text: E.g. LRP5KO +WNT3a – Source file: 26.967; Plot: 26.82; Text: 26.66.

4. p5 lines 28-29: '…, cell lines were cultured as single cells at low density…' This sounds equivocal, since single ESCs do proliferate and form colonies.

5. p5 lines 23-29: You describe that the KO cell lines can be maintained in 2i conditions, but the experiments are performed in serum conditions w/o 2i. Therefore, I was wondering, whether the KO cell lines can be maintained in serum conditions?

6. Fig 1D, E and videos shows protrusions of ESCs: I was wondering, whether the coating of the dish affects cytoneme formation (I believe your Methods section does not indicate, which coating you used)?

7. Length of cytonemes: Since there are references to a figure and videos, I first thought that this is newly generated data for this manuscript. But it seems that this is data from the PNAS paper. Therefore, I suggest to keep the description of the results very brief or move it to the introduction.

8. Videos: I was able to watch all but one (LRP5KO ESC.avi) of the video files. Not sure whether this is related to my hardware but the other videos worked w/o issues.

9. Figure 1F: How did you treat dividing cells in your quantifications? Also: Figure legend indicates n≥3 experiments; N≥50 cells each. However, the source data shows measurements for each condition/genotype for 50 or less than 50 cells (so it is not equal or greater than 50). Could you please clarify whether the plot is based on three biological independent replicates?

10. Figure 2: As mentioned above – how did you treat dividing cells in your quantifications?

11. p.7 line 8: WT cells show a rise (cells with beads) from 20.68% to 50.60% in the first 4 hours. The value 50.6% corresponds to the 3h timepoint in the source data (and not 4h, which is 56.7%).

12. Figure 2D: source data shows values for 50 or more cells per condition/genotype except for CNQX treated cells (n=46) – please modify your figure legend accordingly.

13. Since all imaging measurements are central to this study, I suggest to include a more comprehensive description in the Methods section describing how this was done. If you developed macros for Fiji, these should be included.

14. Figure 3E: Again, less than 50 data points in the source data (49). Please change figure legend accordingly.

15. p.8 line11 – header: Lrp5 and Lrp6 have 'vital' roles in the positioning of Wnt3a beads: The data does not show that the observed effects on bead recruitment and retention is biologically relevant (ie vital)

16. Figure 4: Numbers in source data do not always match with numbers in plots. E.g. WT+iWnt3a: 102 values in total. Plot depicts: n=105. Total number used for calculations for Chi-square test: 101. Another example: WT+Wnt3a beads: total number of values in source data: 100; Plot depicts: n=101; Total number used for calculations for Chi-square test: 102 etc. Please revise this plot and recalculate statistical tests.

17. Referencing of figures in the text should be in consecutive order (which is not the case for Figure 5)

18. Figure 5: Rex1 is a transcription factor, which is localised in the nucleus. However, in Figure 5A it looks like Rex1 is localised in the cytoplasm of cells?

Also, in serum conditions, pluripotency marker expression is heterogeneous. Therefore, I would be interested to see this analysis with the control that you used in all previous experiments, i.e. WT+iWNT3a-beads.

The analysis in Figure 5 does not determine, whether the symmetric or asymmetric distribution of pluripotency markers is preceded by a WNT3a-bead oriented asymmetric division. Therefore, I believe that it is not valid to draw the conclusion that if ESCs show proximal distribution of pluripotency markers, that they underwent WNT-oriented ACD.

I wonder whether KO ESCs or ESCs treated with CNQX show impaired exit from pluripotency and differentiation towards a particular cell lineages?

[Editors’ note: further revisions were suggested prior to acceptance, as described below.]

Thank you for choosing to send your work entitled "Wnt-and Glutamate-receptors orchestrate stem cell dynamics and asymmetric cell division" for consideration at *eLife*. Your letter of appeal has been considered by a Senior Editor and a Reviewing editor, and we are prepared to consider a revised submission with no guarantees of acceptance.

In your point-by-point response to the reviewers' comments you have already clarified some issues and explained how you are planning to address their concerns. In principle we are happy with your suggestions. Nevertheless, we would like to point out essential editorial changes and experiments that should be included in your revised manuscript:

1. Please clarify the relationship between Wnt-mediated cytoneme formation and ACD/cell fate, and the novelty of the data presented in this manuscript and those published in your previous PNAS paper. Furthermore, include additional novel data with knock-out ESC-lines for AMPA or Kainate iGliRs (GriA3/4 and Grik1/3) addressing their role in cytoneme dynamics, Wnt-bead recruitment, spindle orientation etc. as shown in your preliminary data figures.

2. As suggested by reviewer #2 include experimental data performed with additional clones for the respective KO-lines to prove the data shown do not represent artefacts or clonal effects.

3. Please provide more comprehensive descriptions of how the experiments were performed and analysed, and carefully revise numbers in the text, figure legends and the source data file as there have been discrepancies in the reported values.

---

## [Author Response]

[Editors’ note: The authors appealed the original decision. What follows is the authors’ response to the first round of review.]

Reviewer #1:[…] The authors have dissected the role of the Wnt pathway components and iGluRs in cytoneme formation and also in ACD/cell fate. What is not clear is whether the 2 processes are interlinked. Can the authors establish this relationship?

We thank reviewer 1 for understanding the significant advances presented over the PNAS paper, and the novelty in the roles of AMPA/Kainate-type iGluRs and Wnt-pathway components at different stages of Wnt interaction.

The interlinking of the two processes is an interesting point.

We previously showed that soluble Wnt proteins added globally to the media reaches the cell in a cytoneme-independent mechanism and promotes symmetric division and self-renewal instead. In vivo, Wnt proteins are often locally secreted and presented to only one side of the responsive cell.

Our data demonstrates that cytonemes are the mechanism that ESCs use to recruit the localised Wnt ligands. Therefore, without recruitment, the subsequent processes of Wnt-oriented ACD cannot occur.

Our data suggests that Wnt-pathway components are involved in the formation of cytonemes prior to Wnt-contact, whereas the subsequent Wnt-recruitment and dynamics on the membrane are complex interactions between the AMPA/Kainate receptors, the Wnt-pathway, and the cytoskeleton. This dynamic process links initial cytoneme formation to the spatial control of Wnt3a ligands during orientated ACD, which is orchestrated by the activity of both AMPA/Kainate receptor and Wnt pathways. We have expanded the Discussion section to clarify this point.

Reviewer #2:In their manuscript entitled "Wnt-and Glutamate-receptors orchestrate stem cell dynamics and asymmetric cell division" Junyent et al. report a live cell imaging based approach to delineate the contribution of Lrp5, Lrp6, β catenin and iGluR in Wnt mediated asymmetric cell division. Experiments reported in this manuscript appear technically sound and experiments in Figures 4 and 5 provide interesting aspects about Wnt mediated asymmetric cell division in stem cells. However, I have some concerns that substantially diminish my enthusiasm for this paper and feel that the advance in our understanding of asymmetric cell division in stem cells provided by this manuscript is quite limited:1. My main point of criticism is that it is not made clear which aspects and conclusions of this manuscript are substantially novel over the PNAS paper published earlier this year by the same group (Junyent et al., 2020 PNAS). Specifically, data in Figures 1 and 2 report data very similar to Figure 4 of the PNAS manuscript. The individual contributions and differential roles of Lrp5 and Lrp6, the involvement of β-catenin and the effects of CNQX treatment have also been reported in the PNAS paper. This manuscript under review feels like an annex to the PNAS paper rather than a stand-alone story. The authors should take care not to generate the appearance of publishing duplicate findings and focus only on novel data and novel key conclusions.

We would like to thank the Reviewer for this comment and appreciate the concern. In the revised manuscript, we provide additional new data and have reorganised the manuscript to highlight the difference between the novel data of our current manuscript and the published findings in PNAS.

Validation of the Wnt-pathway activation of cell lines, including additional clones by soluble Wnt ligands is carried out in the same way as, but importantly is independent from, previously published data (Junyent et al. 2020). We agree that such data should take a more minor position, and as such this is clearly reflected in the new version of the text.

Figure 2 onwards presents novel data exploring the kinetics of cytonemes and the dynamics of Wnt to cell interactions.

Clone #1 for KO ESCs lacking Wnt-pathway components were featured in our previous publication only in terms of the characteristics and Wnt-reactivity at the individual cytoneme level.

Incorporating suggestions raised at the initial review stage, the new manuscript includes additional KOs for the AMPA and Kainate receptors, and a second cell line for all clonal KO ESCs to address the effects of clonal variation. For each of these cell lines, we have analysed from the pre-Wnt-contact stage of cytoneme formation, through Wnt-membrane dynamics, to the later effects on cell division and cell fate. Previous data in the PNAS paper focused on the initial cell-Wnt interaction. Therefore, the two manuscripts explore contiguous, but independent, biological questions.

The AMPA and Kainate receptor KO cell lines have allowed the dissection of their respective roles throughout Wnt-interaction, and not just at the moment of Wnt3a-bead contact, as in the PNAS paper. We have identified clear differences in their function, particularly in cell fate determination at Wnt-mediated ACD. This, alongside AMPA-specific pharmacological inhibition raises the standard of the study above that of solely CNQX-mediated inhibition of AMPA/Kainate receptors together.

Similarly, the differential roles of Lrp5 and Lrp6 in Wnt interaction were only investigated at a single timepoint in the PNAS paper. Here, we have characterised differences throughout the time course of Wnt-interaction. Of particular note are the findings that Lrp5 KO enhances cytoneme formation higher than that of wildtype, and that Lrp6, but not Lrp5, is required for asymmetric partitioning of cell fate determinants and markers.

Clarification of these novel findings and reducing the involvement of previously published data in the updated manuscript has significantly enhanced the new findings of the manuscript and the understanding of Wnt-ESC interaction, dynamics and ACD.

2. Figure 5 shows that the previously reported deficiency in establishing asymmetric cell division in Lrp5, Lrp6 and β-catenin KOs and CNQX treated cells has direct consequences on the distribution of Nanog between daughter cells.

It is important to note that the involvement of the Wnt/β-catenin pathway components and AMPA/Kainate receptors in spindle orientation and asymmetric cell division (ACD) of ESCs is shown for the first time in this manuscript. We have further clarified this distinction in the revised text.

One aspect that is surprising tough is the asymmetric distribution of Rex1. Rex1 is to my knowledge a non-functional component of the pluripotency circuit (unlike Nanog) and therefore a rather surprising choice for experimentally addressing its asymmetric distribution. I am wondering what could be the developmental rational for such an asymmetric distribution of what seems purely a naïve pluripotency marker, rather than a constituent of the naïve network. Rex1 staining appears largely cytoplasmic in the proximal and distributed panels in Figure 5A, which is surprising for a transcription factor. This should be clarified/commented on. Have the authors assessed the distribution of more functional components of the naive GRN?

We appreciate the reviewer’s comment, and we present the following information to explain the use of Rex1 analysis. Briefly, our previous work (Habib et al., 2013) has demonstrated the ACD of several pluripotency markers, including Nanog, Rex1, Sox2 and Stella, by both live-imaging and protein immunostaining. This work suggested Nanog and Rex1 as robust markers for ACD, while other core pluripotency markers such as OCT4, which is also expressed in epiblast stem cells, is distributed similarly between both Wnt-daughter cells.

Furthermore, Rex1 is widely used as a sensitive reporter for the exit from naïve pluripotency for example in Kalkan et al., 2017, PMID: 28174249, Wray et al., 2011, PMID: 21685889, or Tischler et al., 2019, PMID: 30257965. Rex1, while indeed a transcription factor, can be found in nuclear, perinuclear, and cytoplasmic localizations (e.g. PMID: 22897771, PMID: 21530438) depending on the timepoint.

The asymmetric distribution of both Nanog and Rex1 together have influenced our conclusions of the asymmetric distribution of cell fate markers, and hence Wnt-induced ACD. Additionally, following the reviewer advice and to supplement our data, we have included analysis of Sox2 asymmetry in the updated manuscript, which corroborates the findings with Rex1 and Nanog.

3) Experiments on KO ESCs without performing rescues (by restoration of the endogenous locus or by expression of the missing gene from a transgene) carry the risk of studying artefacts or clonal effects. Regardless of whether KO ESCs show the expected effect or not, these experiments are essential. Alternatively, multiple KOs, generated using different gRNAs can be used to ascertain causative relationships between genotype and phenotype.

We appreciate the reviewer’s concern. Following their advice, and in order to determine KO effects from clonal effects in all of our cell lines, we have newly generated an additional second clone for Lrp5KO, Lrp6KO, Lrp5/6dKO, GriA3/4dKO and GriK1/3dKO. We present their results alongside clone #1 throughout. Through this methodology, we found that initial clonal variability is present in cytoneme formation and early Wnt dynamics, however the clonal behaviours converge in the downstream effect of spindle orientation and the outcome of Wnt-mediated ACD.

Rescue experiments, by overexpression of target protein can have significant issues with artefacts that preclude their use in delicate studies of pathway activity. Not least due to aggregation, a common outcome in overexpression, which can lead to aberrant pathway activation, even in the absence of Wnt ligands (e.g PMID: 27840971, PMID: 11029007). However, in the revised manuscript, we have also included a rescue experiment to further crystallise the roles of Lrp6. Here, we overexpress Lrp6 in a Wnt coreceptor null (Lrp5/6dKO) cell line. This phenocopies Lrp6-only expression (as in Lrp5KO), recovering ACD but not spindle-orientation, and therefore further confirms our conclusions.

Along these lines, WT cells used in this study (W4) are not the proper control for β catenin inducible KO ESCs. Parental cells before 4OHT treatment must be used as control.

The parental cell line of the conditional β-catenin KO (β-catenin^fl/-^) is derived from W4, the same cell type used for wildtype and all KO cells. Additionally, β-catenin^fl/-^ expresses the same levels of pluripotency markers, has a proliferation rate and low apoptosis, similar to WT W4 ESCs (Junyent S., et al. 2020 Figure S7).

Reviewer #3:The manuscript 'Wnt- and Glutamate-receptors orchestrate stem cell dynamics and asymmetric cell division' by Junyent et al. presents a comprehensive analysis on the role of WNT co-receptors Lrp5 and Lrp6, WNT signalling mediator b-catenin, and ionotropic glutamate receptors (iGluRs) in mouse embryonic stem cells (ESCs). In particular, the authors made use of a previously established approach based on microbead-tethered WNTs to expose ESCs to localised WNT3a to study the dynamics of cytoneme formation, WNT3a bead recruitment and stabilisation, and WNT-oriented asymmetric cell division in live-imaging experiments.Generally, the manuscript is well written and the data is appropriately presented. The manuscript describes a number of observations on the kinetics of WNT-bead interactions in in LRP5, LRP6, LRP5/6dKO and b-catenin KO ESCs as well as ESCs treated with iGluR inhibitor CNQX. However, I am not sure about how conclusive the data is in terms of biological relevance as the only functional assay is the analysis of expression of two pluripotency markers after asymmetric cell division. For example, WT ESCs undergo asymmetric cell division (ACD) predominantly oriented towards the WNT3a beads. In contrast, spindle orientation in ESCs without LRP5, b-catenin or iGluRs activity appears randomized. However, this randomized spindle orientation does not have an impact on the asymmetric distribution of pluripotency markers in LRP5 KO ESCs but in ESCs without iGluRs activity.

We thank the reviewer for their attention to a particularly interesting and novel finding of our manuscript – that accurate spindle orientation towards Wnt3a-bead is not necessary for asymmetric cell division as long the mother cell is still in contact with the localised Wnt, such as in the Lrp5KO. We have further highlighted this point in the revised manuscript to clarify the biological relevance. Importantly, In *Drosophila*, for example, the neuroblast can tolerate certain degrees of spindle misorientation due to disruption in internal polarity cues during early stages of cell division. The cells have mechanisms to overcome this initial misorientation and retain the prospective cell fate of the daughter cells (e.g., PMID: 11117748, PMID: 10591217, PMID: 8779714, PMID: 7566172, PMID: 10591217).

This manuscript is a follow up study to the group's recent paper published in PNAS (Junyent et al. 2020). My concern is that some data seems to be redundant to the results presented in the PNAS paper. This manuscript describes WNT3a responsiveness, number and length of cytonemes, WNT3A-bead interactions in LRP5, LRP6, LRP5/6dKO and b-catenin KO ESC lines. As far as I can see, this was already shown in the PNAS paper. Similar videos submitted for this manuscript showing protrusions in each of the KO cell lines were published alongside the PNAS paper. Having said that, I appreciate that this manuscript provides additional insight and a more comprehensive analysis.

We thank the reviewer for their comments, and we are pleased that they find additional insight and more comprehensive analysis in this manuscript over previous findings.

Data previously published in PNAS forms an important part of the background and context for the experiments in this manuscript. This data is clearly referenced, and now, following the advice of the reviewer, we have reduced the emphasis on this information and present it more concisely.

The dynamic study of Wnt is typically made difficult by technical factors such as the tagging of endogenous protein. Our system overcomes these limitations and permits the investigation of Wnt-interaction in its entirety, and as a result highlights key sub-processes that lead to the Wnt-mediated final outcome of ACD. This concept is relatively little-understood, and as such our manuscript has a key novelty in the dissection of the interaction of cells with Wnt-ligands.

For the revised manuscript, we also add additional novel data and additional clonal analysis that have further delineated the mechanism of Wnt-stem cell dynamics and ACD. We obtained knock outs of the Kainate subunits GriK1 and GriK3, which are necessary for Kainate-type receptor formation (Lerma and Marques, 2013, PMID: 24139035), and are expressed by ESCs (Junyent et al., 2020). Additionally, we knocked out the AMPA subunits GriA3 and GriA4, which are also expressed by ESCs (Junyent et al., 2020) and can form Ca^2+^ conductive AMPA receptors (Gull-Candy, 2006, PMID: 16713244). We have included these novel cell lines throughout our study, alongside Wnt-pathway mutants, and have established that both AMPA and Kainate receptors, while not involved in cytoneme formation, are involved in the Wnt-cell interaction at the membrane, and during spindle orientation. Particularly, AMPA-receptors, but not Kainate receptors, facilitate Wnt-mediated ACD. We strongly believe that this additional data significantly improves our understanding of Wnt-ESC dynamics and ACD.

Another concern is technical: The numbers in the source data do not always match the numbers presented in the plots or mentioned in the text. Generally, these are minor differences (see specific comments), but nevertheless need to be carefully revised. Also, the source data for Figure 5 includes data on Dsh2 KO cells, which is not included in this manuscript.

We would like to specially thank the reviewer for the careful revision and data check. We have addressed all the points, and prepared comprehensive supporting data that reflects the figures and results.

1. The manuscript implies that KO ESC lines were generated for this study (e.g. p1 line 18: 'Through live imaging and gene editing, we show…'; p.5 line 6-: '…we knocked-out (KO) the co-receptors…as previously described.'). However, as far as I can see, these cell lines were generated for the PNAS study (Junyent et al. 2020) and re-used for this study, which needs to be more clearly phrased.

We have amended the text to clarify these points and distinguished them from the new clones prepared for the current study.

2. Figure 1B,C: Were the betaKO cells treated for 3 days with 4-HT before WNT3a addition?

Yes. This point is further clarified in the Materials and methods.

3. Figure 1C: Numbers in source file do not exactly match some numbers in plot or text: E.g. LRP5KO +WNT3a – Source file: 26.967; Plot: 26.82; Text: 26.66.

In the revised manuscript, we confirm that the presented or quoted data is accurately reflected in the source data.

4. p5 lines 28-29: '…, cell lines were cultured as single cells at low density…' This sounds equivocal, since single ESCs do proliferate and form colonies.

The core message is that the cells were *seeded* as single cells for the experiments (live imaging or ACD), to allow analysis prior to cell-cell contact. This word choice is now included to capture this notion.

5. p5 lines 23-29: You describe that the KO cell lines can be maintained in 2i conditions, but the experiments are performed in serum conditions w/o 2i. Therefore, I was wondering, whether the KO cell lines can be maintained in serum conditions?

Advances in characterisation of the cells show significant differentiation, even of WT cells, after prolonged culturing in serum containing media. Our live-imaging experiments are indeed performed in Serum w/o 2i, but only last up to 12 hours, which is too short a period to promote differentiation. Even under differentiation-specific conditions, ESCs require between 2-3 days to show markers of differentiation. Hence, Serum + 2i conditions were typically used to maintain ESCs during prolonged maintenance.

6. Fig1D,E and videos shows protrusions of ESCs: I was wondering, whether the coating of the dish affects cytoneme formation (I believe your Methods section does not indicate, which coating you used)?

In our previous manuscript (Junyent et al., 2020) we have shown that dish coating or media does not affect cytoneme formation. The culture conditions have been clarified in the Materials and methods to reflect that commercial tissue-culture treated plates were used without further coating.

7. Length of cytonemes: Since there are references to a figure and videos, I first thought that this is newly generated data for this manuscript. But it seems that this is data from the PNAS paper. Therefore, I suggest to keep the description of the results very brief or move it to the introduction.

The videos displayed are unique to this manuscript and help to compare the morphology of the KO cells of components of Wnt/β-catenin pathway (similar to data presented in Junyent et al., 2020) with the newly generated KOs of iGluRs. Following the suggestion of the reviewer, we have reduced the emphasis on these videos in the revised manuscript.

8. Videos: I was able to watch all but one (LRP5KO ESC.avi) of the video files. Not sure whether this is related to my hardware but the other videos worked w/o issues.

We have reformatted these videos to .mp4 which should be universally accessible.

9. Figure 1F: How did you treat dividing cells in your quantifications? Also: Figure legend indicates n≥3 experiments; N≥50 cells each. However, the source data shows measurements for each condition/genotype for 50 or less than 50 cells (so it is not equal or greater than 50). Could you please clarify whether the plot is based on three biological independent replicates?

During analysis of the cell MSD, only cells that did not divide within the recorded and tracked time (180 min) were included.

In our updated manuscript, we clarify that the analysis includes ≥ 36 cells from each cell line. This sample size is pooled from n ≥ 3 independent experiments (live imaging performed on separate days with a separate batch of Wnt3a-beads), at approximately similar proportions. In the revised manuscript the source data reflects this, and the n/N values have been updated in figure legends.

10. Figure 2: As mentioned above – how did you treat dividing cells in your quantifications?

For measurement of the percentage of cells with beads: for each time point the total number of cells in the frame and the number of cells with beads was calculated and presented as percentage. Dividing cells before cytokinesis were counted as single cells, and those after were counted as two separate cells.

For the retention of the Wnt-bead by the cell following recruitment, only cells that did not undergo division in the 180 minutes following initial contact were included.

11. p.7 line 8: WT cells show a rise (cells with beads) from 20.68% to 50.60% in the first 4 hours. The value 50.6% corresponds to the 3h timepoint in the source data (and not 4h, which is 56.7%).

We apologise for the error and we have corrected the text in the results.

12. Figure 2D: source data shows values for 50 or more cells per condition/genotype except for CNQX treated cells (n=46) – please modify your figure legend accordingly.

We have now corrected the figure legends to accurately reflect the sample sizes.

13. Since all imaging measurements are central to this study, I suggest to include a more comprehensive description in the Methods section describing how this was done. If you developed macros for Fiji, these should be included.

We have expanded our descriptions of the measurement methodology, for all analyses of live imaging data. As indicated in the Materials and methods, all processing and analysis was performed manually in Fiji (ImageJ), hence no macros were used.

14. Figure 3E: Again, less than 50 data points in the source data (49). Please change figure legend accordingly.

We have now corrected the figure legends to accurately reflect the sample sizes.

15. p.8 line11 – header: Lrp5 and Lrp6 have 'vital' roles in the positioning of Wnt3a beads: The data does not show that the observed effects on bead recruitment and retention is biologically relevant (ie vital)

Following the reviewer's suggestion, we have changed the word choice here for clarity.

16. Figure 4: Numbers in source data do not always match with numbers in plots. E.g. WT+iWnt3a: 102 values in total. Plot depicts: n=105. Total number used for calculations for Chi-square test: 101. Another example: WT+Wnt3a beads: total number of values in source data: 100; Plot depicts: n=101; Total number used for calculations for Chi-square test: 102 etc. Please revise this plot and recalculate statistical tests.

In the revised manuscript, the quoted sample sizes are consistent between figure and source data.

17. Referencing of figures in the text should be in consecutive order (which is not the case for Figure 5)

We have reorganized both figures and text. Figures are now referenced in a consecutive and logical order.

18. Figure 5: Rex1 is a transcription factor, which is localised in the nucleus. However, in Figure 5A it looks like Rex1 is localised in the cytoplasm of cells?

We would like to point Reviewer 3 to our response to Reviewer 2 regarding Rex1 as a marker for naïve pluripotency exit (please see above).

Also, in serum conditions, pluripotency marker expression is heterogeneous. Therefore, I would be interested to see this analysis with the control that you used in all previous experiments, i.e. WT+iWNT3a-beads.

We have extensively demonstrated this concept in previous publications (Habib et al., Science 2013) by combining live-cell imaging and protein immunostaining for cell fate markers. The iWnt3a-bead control can be found in Figure S6 of Habib et al. 2013, Science.

The analysis in Figure 5 does not determine, whether the symmetric or asymmetric distribution of pluripotency markers is preceded by a WNT3a-bead oriented asymmetric division. Therefore, I believe that it is not valid to draw the conclusion that if ESCs show proximal distribution of pluripotency markers, that they underwent WNT-oriented ACD.I wonder whether KO ESCs or ESCs treated with CNQX show impaired exit from pluripotency and differentiation towards a particular cell lineages?

As above, this concept has been extensively demonstrated in previous publications (Habib et al. 2013, Science ), also by combining live-cell imaging and protein immunostaining for cell fate markers. Importantly, the Wnt3a-bead-induced asymmetry of the markers was observed during division by live imaging of fluorescent reporters of Rex1, Nanog and Sox2. The iWnt3a-bead does not have this effect.

[Editors’ note: what follows is the authors’ response to the second round of review.]

In your point-by-point response to the reviewers' comments you have already clarified some issues and explained how you are planning to address their concerns. In principle we are happy with your suggestions. Nevertheless, we would like to point out essential editorial changes and experiments that should be included in your revised manuscript:1. Please clarify the relationship between Wnt-mediated cytoneme formation and ACD/cell fate, and the novelty of the data presented in this manuscript and those published in your previous PNAS paper. Furthermore, include additional novel data with knock-out ESC-lines for AMPA or Kainate iGliRs (GriA3/4 and Grik1/3) addressing their role in cytoneme dynamics, Wnt-bead recruitment, spindle orientation etc. as shown in your preliminary data figures.

First, we have reorganized the manuscript, thereby clarifying the novelty of our findings and, in particular, distinguishing new from previously published data. We have now reinforced the connection between cytoneme-mediated Wnt3a-recruitment and asymmetric cell division through the dynamic analysis of Wnt3a–ESC interactions.

We provide novel characterization of ESC lines knocked-out (KO) for the AMPA-type (GriA3/4dKO) and Kainate-type (GriK1/3dKO) ionotropic glutamate-receptors. We have analysed these cells at the level of cytoneme formation and throughout the Wnt3a-interaction, showing that both AMPA and Kainate receptors play roles in the dynamic interaction between stem cells and Wnt3a. At ACD, we determine that AMPA and Kainate receptors play disparate roles in cell fate determination.

2. As suggested by reviewer #2 include experimental data performed with additional clones for the respective KO-lines to prove the data shown do not represent artefacts or clonal effects.

Second, we have generated and analysed a second clone for all Wnt-pathway KO ESC lines included in the previous version, and for all newly-included AMPA and Kainate receptor KOs. We show that, despite minor clonal variability in their initial morphology and early Wnt3a-interaction, their behaviour converges at the downstream effects of spindle orientation and subsequent cell fate determination.

3. Please provide more comprehensive descriptions of how the experiments were performed and analysed, and carefully revise numbers in the text, figure legends and the source data file as there have been discrepancies in the reported values.

Third, we have further expanded the methodology and data analysis details, providing comprehensive source data files to ensure numerical clarity throughout.